# Daytime along-valley winds in the Himalayas as simulated by the WRF–model

Johannes Mikkola[1], Victoria Sinclair[1], Marja Bister[1], and Federico Bianchi[1]

[1]Institute for Atmospheric and Earth System Research, University of Helsinki, Helsinki, Finland.

**Correspondence:** Johannes Mikkola (johannes.mikkola@helsinki.fi)

**Abstract.**

Local valley winds in four major valleys on the southern slope of the Nepal Himalayas are studied by means of high resolution meteorological modelling. The Weather Research and Forecasting model is run with a 1 km horizontal grid spacing covering a 5-day period in December 2014. Model evaluation against meteorological observations from three automatic weather stations in the Khumbu valley (one of the four valleys) shows a good agreement between the modelled and observed daily cycle of the near-surface wind speed and direction. Well defined daytime up-valley winds are found in all of the four valleys during this 5-day period. The night-time along-valley winds are weak in magnitude and flow mostly in the up-valley direction. Differences in the daytime up-valley winds are found between the valleys and their parts. Since the valleys are under similar large-scale forcing, the differences are assumed to be due to differences in the valley topographies. Parts of the valleys with steep valley floor inclination (2–5 degrees) are associated with weaker and shallower daytime up-valley winds compared to the parts which have nearly flat valley floors (<1 degrees). In the four valleys, the ridge heights also increase along the valley, meaning that the valley floor inclination does not necessarily lead to a reduction in the volume of the valley atmosphere. This way the dominant driving mechanism of the along-valley winds, within the valleys, could shift from the valley volume effect to buoyant forcing due to the inclination. Two of the valleys have a 1 km high barrier in their entrances between the valley and the plain. Winds at the valley entrances of these two valleys are weaker when comparing to the open valley entrances. Strong and shallow winds, resembling down-slope winds, are found on the lee-side slope of the barrier followed by weaker and deeper winds at the valley entrance, 20 km towards the valley from the barrier. Although the results of this study are based on a 5–day simulation, they are likely representative of the post-Monsoon season in the study region.

# 1 Introduction

Day-to-day weather in mountain valleys is affected by thermally driven local winds that commonly form under clear skies. The formation of these winds are sensitive to, for example, any large-scale forcing (Whiteman and Doran, 1993) and the geometry of the valley topography (Wagner et al., 2015) which makes them unique for each and every valley. Plain-to-valley winds, and further along-valley winds, stem from the uneven warming of the valley atmosphere and the air above the adjacent plain. The temperature difference causing the along-valley winds is explained by the valley volume effect (Whiteman, 2000). For the

same horizontal area above the valley and above the plain, the air volume in the valley is smaller that above the plain. With the same given solar heating, the valley atmosphere will warm more than the air above the plain which creates a pressure gradient between the plain and the valley. The daytime cross-valley winds, on the other hand, are driven by the buoyancy force that arises from the horizontal temperature difference between the air immediately adjacent to the slopes, which is heated, and the air on the same horizontal level but away from the slope.

In real valley atmospheres, the mechanisms driving along and cross-valley winds together lead to a three-dimensional valley circulation, where during the daytime, the air flows up the slopes and valleys and from the plain into the valley. The daytime cross-valley circulation consist of up-slope winds in the near-surface layer and subsidence in the valley atmosphere away from the slopes. Subsidence warming is the dominant mechanisms leading to the heating of the air in the core of the valley in the morning transition phase whereas the turbulent convective heat flux from the valley floor and the slopes is the dominant

mechanism in the afternoon (Serafin and Zardi, 2010). Similarly, the up-slope winds typically form just after sunrise, peaking before noon, whereas the up-valley winds develop later during the day and peak in the afternoon (Whiteman, 2000). Although in the cross-valley circulation the subsiding air causes local warming in the core of the valley atmosphere, the net effect of the cross-valley circulation is to export heat out of the valley atmosphere due to the overshooting up-slope winds at the valley crests (Schmidli, 2013). Due to the heat export, the valley volume effect is considered as the theoretical maximum of the heating of

the valley atmosphere compared to adjacent plain.

Accurate modelling of these thermally driven winds requires high horizontal resolution, down to at least 1 km grid spacing, due to their complex structure and sensitivity to the topography (Schmidli et al., 2018). The local valley circulation, and vertical exchange of heat and momentum between the valley and the free troposphere, is inaccurately modelled in climate models that are typically run at coarse resolution up to 100–200 km (Rotach et al., 2014). With coarse resolution models, it is also not

possible to simulate accurately the vertical transport of aerosol, emitted in the mountain boundary layer and ventilated by the valley winds into the free troposphere. Once in the free troposphere, aerosol are less subject to removal processes, can undergo chemical transformation and long-range transport in which way the ventilated aerosol can effect areas remote from the actual valley in which they formed. Since over half of the Earth's land surface is considered as complex terrain (Rotach et al., 2014), this creates a high uncertainty e.g. in the global carbon budget and in climate change predictions.

This study concentrates on four valleys located in the Nepal Himalayas during a 5-day period in December 2014. The local wind patterns in the Khumbu valley (one of the valleys that is investigated in this study) have been studied in the past by means of meteorological observations (Inoue, 1976; Ueno and Kayastha, 2001; Bollasina et al., 2002; Ueno et al., 2008; Bonasoni

et al., 2010; Shea et al., 2015; Yang et al., 2018) and high-resolution meteorological modelling (Karki et al., 2017; Potter et al., 2018, 2021). Overall, the daily cycle of the along-valley winds in the Khumbu valley are similar between these studies with well defined daytime up-valley winds and weaker night-time winds flowing either in the up or down-valley direction. A recent study showed that the Khumbu valley could act as a source of pre-industrial aerosol in the free troposphere – model results indicate that the biogenic vapors emitted in the Khumbu valley were oxidized and then transported to free troposphere by the daytime up-valley winds (Bianchi et al., 2021). They suggest that other similar valleys on the southern slope of the Himalayas would likewise form and transport biogenic aerosol to the free troposphere. However, the local valley wind systems in the other major valleys located nearby in this region have not yet been well studied. Therefore, it remains unknown if these other major valleys have similar along-valley winds as found in the Khumbu valley and thus if they could also act as sources of free tropospheric aerosol.

This study focuses on along-valley winds during a carefully selected 5-day period in December 2014 that is representative of the post-Monsoon season. The first aim of this article is to identify the characteristics (wind speed, depth of the flow, diurnal cycle) of the local valley wind system in four major valleys in the Hindu-Kush Karakoram Himalayan (HKKH) region during this 5-day period in December 2014. The second aim is to identify notable differences between the four valleys in terms of their local wind systems. The third aim is to attempt to explain the causes for the differences in the local winds. These aims are addressed primarily by analysing a 5-day simulation performed with the Weather Research and Forecasting (WRF) model as the network of meteorological observations is rather scarce in this region.

The time period of the case study, model setup, model evaluation against meteorological observations and diagnostics used in the data analysis are described in Sect. 2. The valley topography characteristics are described in Sect. 3. The along and cross-valley wind characteristics in each of the four valleys are described in Sect. 4. The differences in the valley winds between the valleys are discussed in Sect. 5 and the conclusions are given in Sect. 6.

## 2 Methods

 ### 2.1 Time period of the case study

Due to computational and data storage limitations, it is not possible to perform a long-term simulation with high temporal resolution output. Therefore, in this study we do not attempt to provide a climatology of the thermally driven winds. Instead, we select a 5-day period to simulate and perform an in-depth case study.

Bollasina et al. (2002) describes the seasonal variation and climatology of the study area by means of 6 years of meteorological observations made at the Nepal Climate Observatory – Pyramid station (NCO–P) located close to the base of Mount Everest (marked as a black star in Fig. 2b). Generally clear skies and maximum in daily temperature ranges are observed in Decembers 1994-1999 at NCO-P (Bollasina et al., 2002). Similarly, during 2006-2007 winter seasons (NDJF) clear skies and strong diurnal variation in temperature were observed by Bonasoni et al. (2010). Based on the observations at NCO-P, December is a good period for studying the thermally driven mountain winds in this region.

The period of this case study was selected to coincided with the measurement campaign by Bianchi et al. (2021) that took place in the Khumbu valley between 29 November - 25 December 2014 as additional observations and analysis of the local weather conditions were already available. Five consecutive days with clear skies were selected for the study period to simulate the thermally driven along-valley winds in this region. During 17-21 December 2014 weather at the measurement site was reported to be sunny (Supplementary material Bianchi et al. (2021)). Comparing the observed wind direction during the study period 17-21 December 2014 (Bianchi et al., 2021) with other periods such as winters 1994-1999 (Bollasina et al., 2002; Ueno and Kayastha, 2001), December 2003 (Ueno et al., 2008) and Decembers 2006-2007 (Bonasoni et al., 2010), the study period resembles well the average wind conditions at the NCO-P station.

To determine if the large-scale flow during our 5-day period is representative of the long-term climatology, we compare daily averages of upper-level winds from the study period to the 40-year climatology using ERA5-reanalysis (Hersbach et al., 2020). The large-scale flow is examined at 400 hPa, since at many locations within the study area the surface pressure is below 500 hPa. The 40-year climatology considered is the average of Decembers 1980-2019. The 40-year average and the daily averages of the large-scale flow during the study period are shown in Fig. 1. The border of Nepal is shown in red in Fig. 1 and the study area is located in the north-eastern part of Nepal. The 40-year December average of the 400-hPa wind is mostly westerly with the sub-tropical jet located on average around 30 degrees north. The sub-tropical jet is located around 25 degrees north during 17-21 December 2014 and south-east of the study area. The jet seen in the 40-year average (Fig. 1a) is much wider than in any of the daily averages which suggests the exact latitudinal position of the jet does vary from day-to-day. Although the jet is located around 5 degrees south of the 40-year average during our study period, the relatively large standard deviation implies the study period is still representative of the areas climatological large-scale wind condition. During 17-21 December 2014, the daily averaged wind speeds differ from the 40-year average by less than 1 standard deviation (SD) of the December monthly means near the study area (Figures 1b-f). During 17 December 2014 the large-scale wind direction is anomalous of the study area's climatology, since the flow is northerly around eastern Nepal (Fig. 1b). Therefore, the 17th December 2014 will be not considered in the analysis of the local valley winds in the study area due to its' anomaly compared the other days of the study

period. During the remainder of the study period, the large-scale wind speed and direction is not anomalous of the study area's long-term climatology.

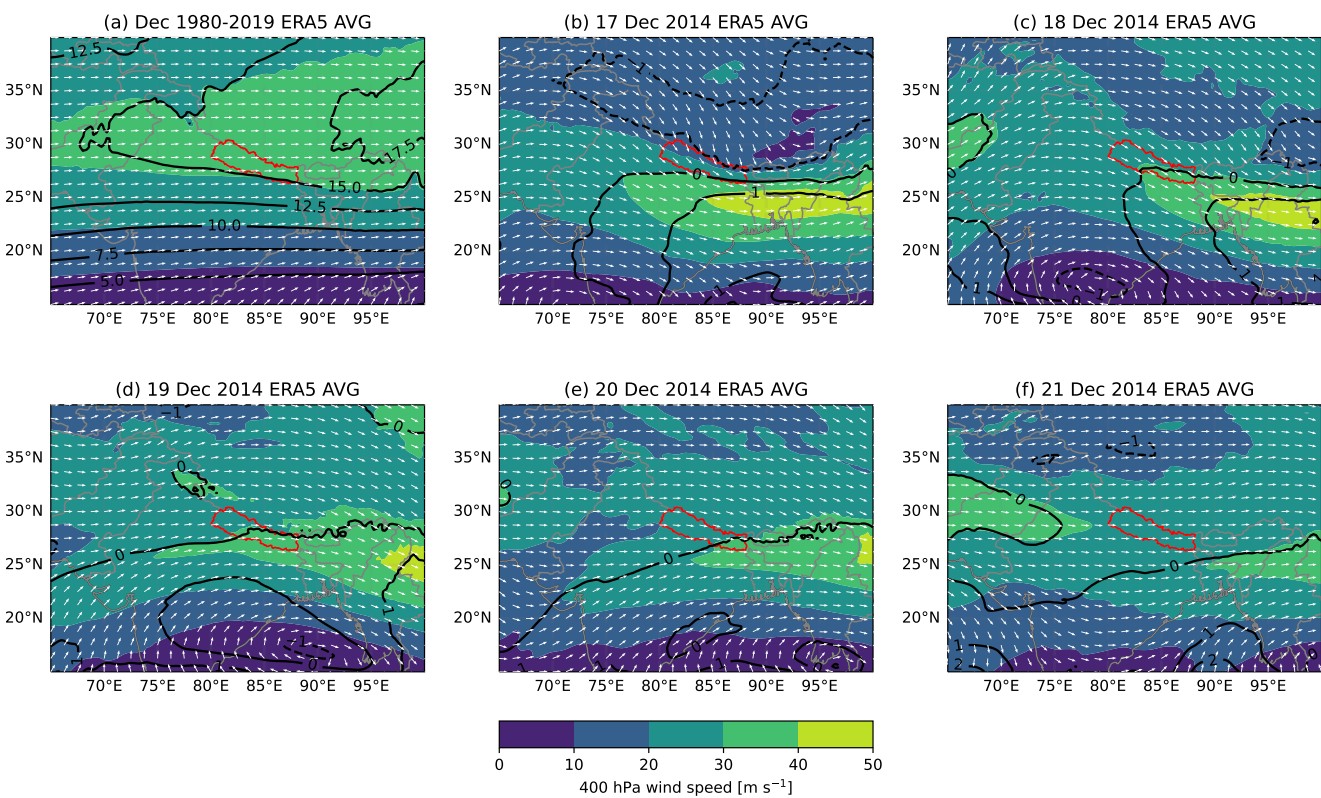

**Figure 1. (a)** 400-hPa wind speed (shading) and direction (white vectors) averaged over Decembers 1980-2019 in ERA5 reanalysis. Standard deviation of the 40-year monthly means is shown by the black contours (m s$^{-1}$). **(b)-(f)** Daily averaged wind speed and direction at 400 hPa in ERA5 reanalysis for 17-21st Dec 2014. Standard score of the daily average wind speed compared to the 40 year monthly means is shown by the black contours with a contour interval of 1 SD. Coastlines and country borders are shown in grey and borders of Nepal are shown in red.

## 2.2 Weather Research and Forecasting model setup

The Weather Research and Forecasting model (WRF) is a state-of-art numerical weather prediction model that is widely used both in operational and research purposes. The version used in this study is WRF 3.6.1.

The simulation used in this study is identical to what Bianchi et al. (2021) analysed in their study. The simulation consists of four nested domains (referred to as d01 (outer domain), d02, d03 and d04 (inner domain)) that are illustrated in Fig. 2a. The outer domain covers an area of 3618 by 2997 km and was run with a horizontal grid spacing of 27 km. The horizontal grid

spacing decreases to 9 km (d02), 3 km (d03) and finally to 1 km in the innermost domain that covers an area of 288 by 300 km. All of the domains are run with 61 vertical levels.

The simulation was initialised using the Climate Forecast System Reanalysis (Saha et al., 2010) which has a horizontal grid spacing of 0.5°. The simulation was kept on track by nudging the temperature, horizontal wind components and specific humidity in the outer domain (d01) towards the reanalysis every 6 hours. The nudging was only performed above the atmospheric boundary layer. The surface topography is from United States Geological Survey with horizontal resolution of 30 arc seconds ($\sim$ 1 km) and is shown for domains d01 and d04 in Figures 2a and 2b, respectively. An adaptive time-step, with a target Courant-Friedrichs-Lewy value of 0.8, was used to keep the simulation numerically stable. For the inner domain (d04) this means a time-step of approximately 1 second. In addition, sixth order numerical diffusion and w-Rayleigh damping was applied to the uppermost 5 km. Sub-grid scale processes are parameterised as follows: the Thompson scheme for microphysics (Thompson et al., 2008), the RRTMG scheme for long-wave and short-wave radiation (Iacono et al., 2008), the Mellor-Yamada-Janjic (Eta) TKE scheme for boundary-layer turbulence and the Eta Similarity scheme for the surface layer (Janjic, 1994). The land-surface scheme used in the simulation was the Unified Noah Land Surface Model (Tewari et al., 2004). Convection was parameterised based on Kain-Fritsch scheme (Kain, 2004) but only in domains d01 and d02.

The model was run as one continuous five-day simulation initialised at 00:00 UTC (5:45 local time) on the 17th December 2014 and ran until 23:59UTC on the 21st December 2014. The output frequency of the simulation was one hour (d01), 30 minutes (d02), 10 minutes (d03) and 5 minutes (d04). In general, numerical weather prediction models require a spin-up period to allow the model to adjust from the specified initial conditions to a balanced state which is consistent with the models' own dynamics and physics. Bonekamp et al. (2018) studied the effect of spin-up time on the output from high-resolution WRF simulations in the Langtang catchment, which is located just outside our innermost WRF domain to the west. They found that the difference between 12 hours, 24 hours, and 3 days spin-up time was relatively small (10-m wind Root Mean Square Error 3.5 m s$^{-1}$ for 12 hour and 3.3 m s$^{-1}$ for 24 hour and 3 days). Therefore, we assume a spin-up time of 12 hours for our simulation, and exclude the output from the first 12 hours of the simulation in the analyses but is still included in the plotted timeseries.

## 2.2.1 Model evaluation

According to recent studies comparing high-resolution WRF simulations to observations in the European Alps (Giovannini et al., 2014a, b) and in the Himalayas (Collier and Immerzeel, 2015; Karki et al., 2017; Potter et al., 2018) our model setup is suitable for studying local valley winds within the inner domain in our simulation. However, Collier and Immerzeel (2015) suggested that their WRF simulation with 1 km horizontal grid spacing is not accurate enough to fully resolve the thermally driven valley circulations in the narrowest parts (length scales less than 2 km) of the Langtang catchment, which is located just outside our innermost WRF domain to the west. This would limit the reliability of the model simulation in the smallest sub-branches of the main valleys. The valleys we concentrate on have ridge-to-ridge distances of more than 30 km (discussed in Sect. 3). Singh et al. (2021) found WRF struggled to accurately simulated the diurnal variation in wind speed, especially

during daytime and when wind condition changes from high to low, over the Central Himalaya. However, they simulated the

monsoon season which is not favorable for the thermally-driven valley winds that are the focus of our study.

The WRF simulation is evaluated by comparing the modelled near-surface temperature and winds to meteorological observations from three automatic weather stations (AWS) in the Khumbu valley (Table 1). The Khumbu valley is shown by the second-left yellow line in Fig. 2b. The black star, blue cross, and red cross on the yellow line show the location of the Nepal Climate Observatory – Pyramid station (NCO–P), Namche AWS and Lukla AWS, respectively. To compare the modelled val-

ues to the observations, model variables were taken from the closest model grid point. The grid point was selected based on the station and model grid coordinates without the use of any horizontal interpolation. Further details of the observation sites can be found in Yang et al. (2018) and *http://geonetwork.evk2cnr.org*.

The lowest wind component in the model output is at 10 m above the surface whereas the winds are observed at 5-m height. The modelled wind speed is adjusted to 5 m, assuming a logarithmic wind profile for neutral stratification (Stull, 1988). The

temperatures are both observed and modelled at 2-m height above the surface but the model temperatures are adjusted to correspond to the altitude of the observation site based on the dry adiabatic lapse rate. NCO–P and Namche AWS are located in open areas whereas Lukla AWS is surrounded by trees, according to the station photos in *http://geonetwork.evk2cnr.org*. The threshold wind speed, below which reliable measurements cannot be obtained, for the anemometers measuring at all three stations is 0.21 m s$^{-1}$ for wind speed and 0.15 m s$^{-1}$ for wind direction. Timesteps with wind speeds observed below these

thresholds are neglected in the following comparison.

The model evaluation is based on the Mean Bias Error (MBE), Mean Absolute Error (MAE) and Root Mean Square Error (RMSE) (Inness and Dorling, 2012) of the 2-m temperature (2mT) and 5-m wind speed.

MBE, MAE and RMSE are calculated using Equations 1, 2 and 3, respectively:

$$\text{MBE} = \frac{1}{n}\sum_{t=1}^{n} X_{wrf}(t) - X_{obs}(t) \tag{1}$$

$$\text{MAE} = \frac{1}{n}\sum_{t=1}^{n} |X_{wrf}(t) - X_{obs}(t)| \tag{2}$$

$$\text{RMSE} = \sqrt{\frac{1}{n}\sum_{t=1}^{n} (X_{wrf}(t) - X_{obs}(t))^2} \tag{3}$$

**Table 1.** Station and model grid point locations used in the comparison to observations.

|  | Observation Station | | Model grid point | |
|---|---|---|---|---|
| NCO-P | 27.959N 86.813E | 5050 m.a.s.l | 27.957N 86.810E | 5005 m.a.s.l |
| Namche | 27.802N 86.715E | 3570 m.a.s.l | 27.804N 86.718E | 3360 m.a.s.l |
| Lukla | 27.696N 86.723E | 2660 m.a.s.l | 27.696N 86.728E | 2592 m.a.s.l |

where $X_{wrf}(t)$ and $X_{obs}(t)$ are the modelled value at the closest model grid point and the observation at the station at the same timestep $t$, respectively. The calculated error metrics for the comparison are presented in Table 2. The comparison is separated to daytime (06-18 local time) and night-time (18-06 local time) timesteps. The spin-up time of 12 hours from the beginning of the simulation is excluded. The observations are provided as hourly averages meaning that the comparison includes 48 timesteps for daytime and 60 timesteps for night-time in total. The number of missing timesteps due to neglected wind speed measurements is 2 in Namche night-time, 3 in Lukla daytime and 9 in Lukla night-time. No missing or neglected timesteps in temperature observations or in NCO-P wind observations. The modelled and observed 2mT and 5-m winds are shown as a timeseries in Supplementary Figure A1.

The RMSE in 2mT ranges between 2.5–2.7 K for daytime (06-18LT) and 2.2–3.5 for night-time (18-06LT) (Table 2). In the WRF simulation performed by Bonekamp et al. (2018), which has the same horizontal grid spacing and spin-up time as our simulation, the mean RMSE over all stations in Lang-tang catchment is 8.0 K which is notably larger than in our simulation. At NCO–P and Namche, the temperature in our simulation is underestimated for most of the simulation (MBE is negative for both stations) but the simulated amplitude of the diurnal cycle is similar to the observed amplitude (Supplementary Fig. A1b,d). In Lukla, the amplitude of the daily cycle in temperature is not captured well (MAE 2.3 K) but the average temperature is closer to observations (daytime MBE -0.5 K and night-time MBE 2.3 K). In Lukla, the modelled maximum and minimum temperatures are around 2.5 K above and below the observation respectively. However, the timing of the diurnal cycle is well captured in all of the three stations (Supplementary Fig. A1b,d,f).

The RMSE in 5-meter wind speed ranges between 1.4–3.2 m s$^{-1}$ during daytime and 0.5–2.9 m s$^{-1}$ during night-time (Table 2). In the WRF simulation performed by Bonekamp et al. (2018) WRF-simulation, the mean RMSE over all stations in Lang-tang catchment is 3.5 m s$^{-1}$ which is a similar range as we compute. For both day and night-time, the RMSE in wind speed is lowest at Lukla and highest at NCO-P and, at each station, the night-time RMSE is lower than daytime. At each station the diurnal cycle in wind direction is modelled well, especially the daytime southerlies (Supplementary Figure A1a,c,e). In NCO-P both the observed and modelled near-surface winds do not have a clear diurnal cycle in speed (Supplementary Figure A1a) which is seen also in the skill scores (MAE daytime 2.5 m s$^{-1}$, night-time 2.4 m s$^{-1}$). In Namche, a clearer diurnal cycle is seen both in modelled and observed near-surface wind speeds (Supplementary Figure A1c). At Lukla, the observed wind

**Table 2.** WRF modelled values compared to near-surface observations in the Khumbu valley. Mean Bias Error (MBE), Mean Absolute Error (MAE) and Root Mean Square Error (RMSE) of 2 m temperature and 5 meter wind speed separately for daytime (06–18 local time) and night-time (18–06 local time).

| | 2 meter temperature [K] | | | | | | 5 meter wind speed [m s$^{-1}$] | | | | | |
|---|---|---|---|---|---|---|---|---|---|---|---|---|
| | MBE | | MAE | | RMSE | | MBE | | MAE | | RMSE | |
| | Day | Night | Day | Night | Day | Night | Day | Night | Day | Night | Day | Night |
| NCO-P | -1.9 | -3.2 | 2.4 | 3.2 | 2.7 | 3.5 | -1.7 | 1.1 | 2.5 | 2.4 | 3.2 | 2.9 |
| Namche | -1.9 | -1.5 | 2.1 | 1.9 | 2.5 | 2.2 | 1.4 | 0.6 | 1.7 | 1.0 | 2.1 | 1.6 |
| Lukla | -0.5 | 2.3 | 2.3 | 2.3 | 2.6 | 2.5 | 1.2 | 0.3 | 1.2 | 0.4 | 1.4 | 0.5 |

speeds are less than 2 m s$^{-1}$ during the whole 5 day period and the modelled wind speeds are notably larger and exhibit a

200 clearer diurnal cycle (Supplementary Figure A1e) which leads to higher MAE and MBE during daytime (1.2 m s$^{-1}$) compared to night-time (MAE 0.3 m s$^{-1}$, MBE 0.4 m s$^{-1}$). The weak observed winds may be due to sheltering by the nearby trees and thus the Lukla AWS may not be representative of the area covered by the closest WRF grid box. However, overall the modelled diurnal cycle of winds, in both magnitude and direction, agrees reasonably well with the observations and hence we conclude that the WRF simulation is sufficient in simulating the valley winds.

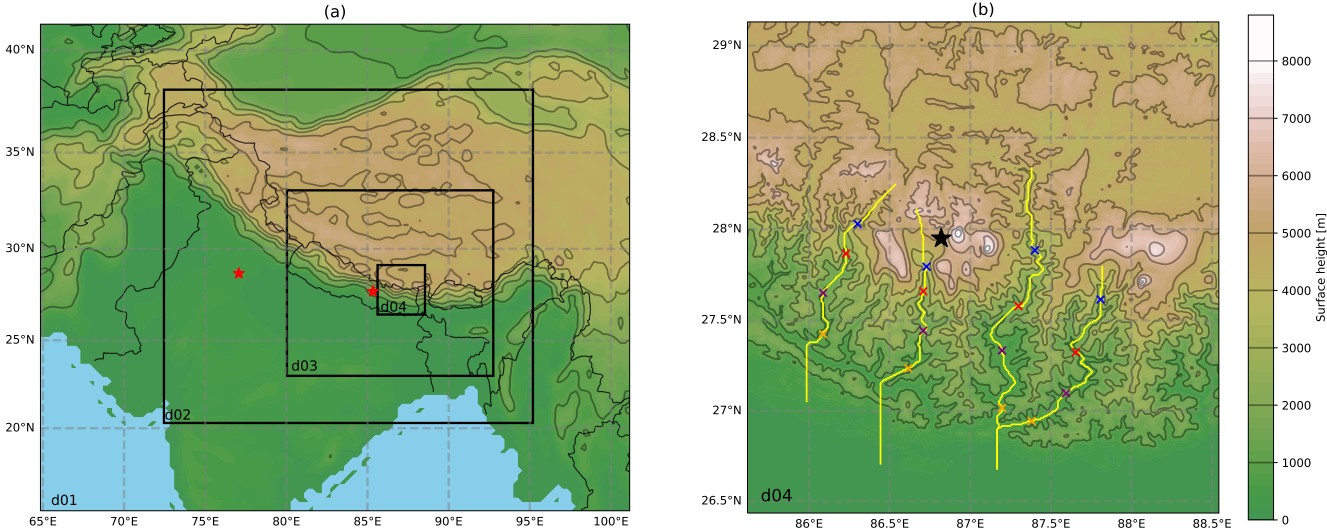

**Figure 2. (a)** Topography of the d01 domain and the inner domains d02, d03 and d04 in the WRF simulation. Thin solid lines denote the country borders. Red stars denote the location of Kathmandu (the furthest east star) and New Delhi. **(b)** Topography of the d04 domain. Yellow lines present the valley center lines used in the analysis. Black star indicates the location of the Nepal Climate Observatory – Pyramid station (NCO–P). Coloured crosses denote locations along the valley center lines that are referred in text and other Figures.

## 2.3 Diagnostics

### 2.3.1 Valley geometry

In this section we define a number of diagnostics which quantify the topography and geometry of the valleys and thus enables us to quantitatively compare the four different valleys.

The valley center lines (in yellow in Fig. 2b) were identified using a simple algorithm. The algorithm starts from the valley top, from a grid point chosen by the user, from where it finds the way down to the valley entrance by choosing the neighbouring grid point with the lowest surface height. This procedure continues over and over until the valley entrance is reached (determined by the user). The yellow lines in Fig. 2b were manually extended from the valley entrance to reach over the adjacent plain. In the case of the two westernmost valleys – the Gaurishankar and Khumbu valleys - the valley center lines were extended due south over the perpendicular barrier from the valley entrance instead of continuing with the algorithm until the plain is reached. The daytime up-valley winds are connected to the plain by winds that propagate over this topographical barrier so it is reasonable to follow this line in the analysis (discussed further in Sect. 4).

The ridge lines (not shown) were identified using a similar algorithm, with a small modification to the aforementioned. Now the algorithm chooses the neighbouring grid point that has the highest surface height but, in addition, is forced to propagate into the assumed direction of the ridge, chosen by the user based on the topography map. The ridge line determination is a difficult task, since the valleys have, for example, a couple of Eight-thousander[1] around them. However the use of these identified ridge lines allows us to approximately estimate the average depth of the valley atmosphere in these valleys.

The valley width is calculated at an elevation of 1000 m above the valley center line. The width estimate is the distance between the west and east wall from the grid points where the elevation is above 1000 m compared to the valley center line height. The west and east walls are found by moving in a direction perpendicular to the valley center line. This value does not give an absolute value for the valley widths, but does provide a comparable number for different parts of each valley, which was a more convenient approach to visualise the narrowing and widening of the valley topographies with the complex valley topographies with multiple side-valleys. The narrowing and widening of the valley topography can be important for the valley volume effect related to the along-valley winds. The elevation of 1000 m above the valley center line was chosen based on the valley depths; most of the valleys have a minimum depth of 1000 m (valley topographies described in details in Sect. 3).

### 2.3.2 Along and cross-valley wind components

Since the valleys are not exactly north-south orientated at every point in the along-valley direction, the meridional and zonal winds in the model output can not be considered as the along and cross-valley wind components directly. The along and cross-valley wind components are the wind components parallel and perpendicular to the valley center line (described in Sect. 2.3.1).

---

[1]Mountain tops reaching above 8000 m in the inner domain (d04) of the simulation; Mt Everest, Kanchanjunga, Lhotse, Makalu, Cho Oyu

We use the vector $\boldsymbol{A}_i = \boldsymbol{x}(x_{i+3} - x_{i-3}) + \boldsymbol{y}(y_{i+3} - y_{i-3})$ to describe the local orientation of the valley at $i$th grid point on the valley center line. Here $(x_i, y_i)$ denote the coordinates of the $i$th grid point on the valley center line in the inner model domain grid. Three grid points before and after the actual location are used to smooth out the sharpest turns in the valley center line. Here we write the horizontal wind in Cartesian coordinates, $\boldsymbol{V} = \boldsymbol{x}u + \boldsymbol{y}v$, where $u$ and $v$ refer to the zonal and meridional wind components. The along-valley wind component at the valley grid point $i$, $AVW_i$, is then calculated using Eq. 4:

$$AVW_i = \frac{\boldsymbol{A}_i}{|\boldsymbol{A}_i|} \cdot \boldsymbol{V}_i = \frac{u_i(x_{i+3} - x_{i-3}) + v_i(y_{i+3} - y_{i-3})}{\sqrt{(x_{i+3} - x_{i-3})^2 + (y_{i+3} - y_{i-3})^2}}. \tag{4}$$

The cross-valley wind component is calculated using the same approach. Here the cross-valley wind is perpendicular to the along-valley wind but it is calculated at the slopes, 5 to 10 grid points away from the valley center line. The grid points on both slopes are selected to have the same elevation gain with respect to the valley center line height and thus can differ in horizontal distance from the center line. The wind component, describing the cross-valley winds at the slopes located around the $i$th grid point in the valley center line is calculated using Eq. 5:

$$CVW_i = \left( \boldsymbol{z} \times \frac{\boldsymbol{A}_i}{|\boldsymbol{A}_i|} \right) \cdot \boldsymbol{V}_i = \frac{-u_i(y_{i+3} - y_{i-3}) + v_i(x_{i+3} - x_{i-3})}{\sqrt{(x_{i+3} - x_{i-3})^2 + (y_{i+3} - y_{i-3})^2}}. \tag{5}$$

## 3 Overview of the topographic characteristics of the four valleys

The four valleys considered in this study, and their center lines identified by the algorithm described in Sect. 2.3.1, are marked with yellow lines in Fig. 2b. The valleys are located along the southern slope of the Nepalese Himalayas and they are called Gaurishankar, Khumbu, Makalu and Kanchanjunga, listed from west to east. The valleys are roughly north-south orientated and are inclined towards the north so the valleys face south. Figure 3 gives an overview of the topographic characteristics along each of the four valleys based on the diagnostics defined in Sect. 2.3.1. All of the valleys have a similar degree of valley narrowing (decrease in valley width) from the valley entrance towards the valley top (blue lines in Fig. 3). The Khumbu valley is an exception as the valley becomes broader (i.e. the valley width increases) from the along-valley grid point 40 (horizontal axis in Fig. 3b), at the valley floor elevation of 3000 m, towards the valley top (along-valley grid point 0). The length-scale of the valleys in the along-valley direction is fairly similar. Gaurishankar is the shortest valley at approximately 80 km whereas Makalu is the longest valley with a horizontal length of approximately 120 km in the along the valley direction from the valley entrance to the valley top (distance along the valley center line from blue to yellow cross).

The valleys can be roughly divided in two groups based on their topographic characteristics: the two westernmost valleys, Gaurishankar and Khumbu (Figures 3a–b), and the two easternmost, Makalu and Kanchanjunga (Figures 3c–d). Makalu and Kanchanjunga have rather flat valley floors from the valley entrance into the valley with less than 1 degrees inclination (brown shading in Fig. 3c–d). After 60 grid points (approximately 60 km) into the valley, the valley floors start to incline towards the valley top. In the Makalu valley, the valley floor elevation increase from 500 to 4000 m.a.s.l from grid point 120 to grid point 30 with the slope varying between 0–6 degrees. In the Kanchanjunga valley, the floor inclination ranges between 1–3 degrees in the top-half of the valley, between the grid points 20 and 60. In contrast, the Gaurishankar and Khumbu valleys both have a

continuous increase in the inclination that increases from 2 degrees (along-valley grid points 80-100) to 5 degrees towards the valley top.

The valley depth (the difference between the ridge and valley center line heights) is lowest in the Gaurishankar valley. The valley depth is shown by the purple line in Fig. 3 with the values on the right-hand side vertical axes. In Gaurishankar, the valley depth reaches around 2500 m whereas the valley depth in the other three valleys is larger and is in the range 3500–4000 m. In the portions of the Makalu and Kanchanjunga valleys where the valley floor is flat, the valley depth increases at a constant rate from the valley entrance towards the valley top. In the Kanchanjunga valley, the approximately constant increase continues until the valley top. In the Gaurishankar and Khumbu valleys, the valley depth increases are not as gradual but still the valleys get deeper from the valley entrance towards the valley top.

The valley entrances are significantly different between the valleys. The up to 1000 m high and 10 km wide topographic barrier in the along-valley direction lies perpendicular to the entrances of the Gaurishankar and Khumbu valleys (around grid point 120 in Figures 3a–b). The barrier has an inclination of up to 10 degrees on both the northern and southern slopes of the barrier. The entrance into the Makalu and Kanchanjunga valleys are, in contrast, open without any such obstacle.

To summarise, the two westernmost valleys (Gaurishankar and Khumbu, Figures 3a–b) have inclined valley floors throughout the length of the valley and an 1 km high perpendicular barrier between the valley entrance and the plain instead of an open entrance, when comparing to the two easternmost valleys (Makalu and Kanchanjunga, Figures 3c–d) that have a 40 km long flat portion into the valley from the open valley entrance.

# 4  Results

We first analysis the large-scale flow to ensure it is consistent with the analysis in ERA5 (discussed in Sect. 2.1). Fig. 4 shows the 400–hPa wind in the outer domain (d01) in the WRF simulation at 12 local time (noon) on 18-21 December 2014 (17 Dec is not considered as it is within the spin-up period). 12 local time is considered since the main focus of the study is the daytime local winds. Overall, the 400–hPa wind speed and direction are similar between our simulation and the daily averages of ERA5 (Sect. 2.1). Most important for the study, the location and wind speed of the 400–hPa subtropical jet are very similar with respect to the high-resolution domain (d04) of our simulation.

The wind speed and direction above the valleys, and particularly at ridge height, can influence the winds within the valleys (Whiteman and Doran, 1993; Solanki et al., 2019; Lai et al., 2021). The large-scale flow above the valleys changed from north-westerlies (18th Dec) and to westerlies (19-21th Dec), which is seen in Fig. 4, where the red rectangular denotes the location of the inner domain d04. The 400-hPa wind speed above the valleys is around 30-40 m s$^{-1}$ during 18-20 Dec (Fig. 4a-c) and weaker during 21 Dec with 20-30 m s$^{-1}$ winds (Fig. 4d). The sub-tropical jet was located south-east of the inner domain (d04) location during the 18–21 Dec simulation period.

Daytime up-valley winds are found in all of the four valleys on each day of the simulation (Fig. 5 - along-valley winds are discussed in detail in Sect. 4.1). The strongest daytime near-surface winds in the valleys are found around the valley center

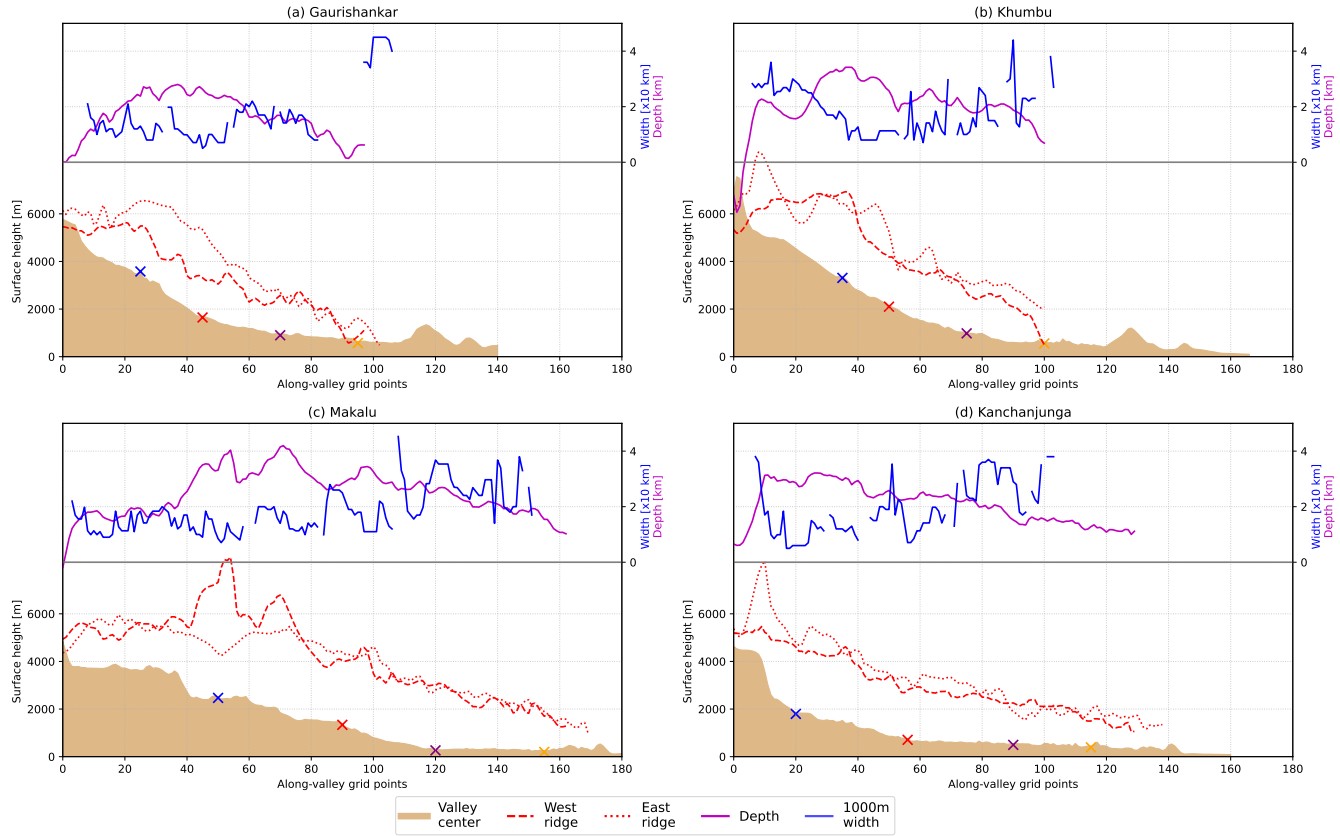

**Figure 3.** Height profiles (brown shading) of the valley center lines (yellow lines in Fig 2b) and the ridge lines (red dashed lines) correspond to the values on the left-hand y-axis. Cross-valley width at an elevation of 1000 m above the valley center line is shown with the blue line. The valley depth is described as the height difference between the center line and average of the ridge lines which is shown in purple. The lines for width and depth correspond to the values on the right-hand y-axis. Each panel shows one valley: (a) the Gaurishankar Valley (most westerly valley), (b) the Khumbu Valley, (c) Makalu and (d) Kanchanjunga.

lines (white dashed lines in Fig. 5) and have magnitudes of up to 10 m s$^{-1}$. The up-valley winds spread also into the smaller valleys branching off from the main valleys, but are weaker than the up-valley winds in the main valleys.

The large-scale north-westerly winds at upper levels channel into the valleys, especially in the northernmost and thus highest parts of the valleys Gaurishankar and Makalu, during the 18th Dec which is seen as strong down-valley near-surface winds during the day (Fig. 5a) and during the night between the 17–18 Dec (Supplementary Figure A2a,c). Up to 25 m s$^{-1}$ near-surface down-valley winds are found in the tops of the valleys where the valley floors are at 4000–5000 m above sea level. The tops of the Gaurishankar and Makalu valleys are favorable for large-scale north-westerlies to channel into the valley atmosphere. The

north-westerlies penetrate into the valley atmosphere through the gaps and open south-north orientated structures surrounding the top of the valley. The easternmost valley, Kanchanjunga, is an exception here since it is surrounded by high-enough ridges

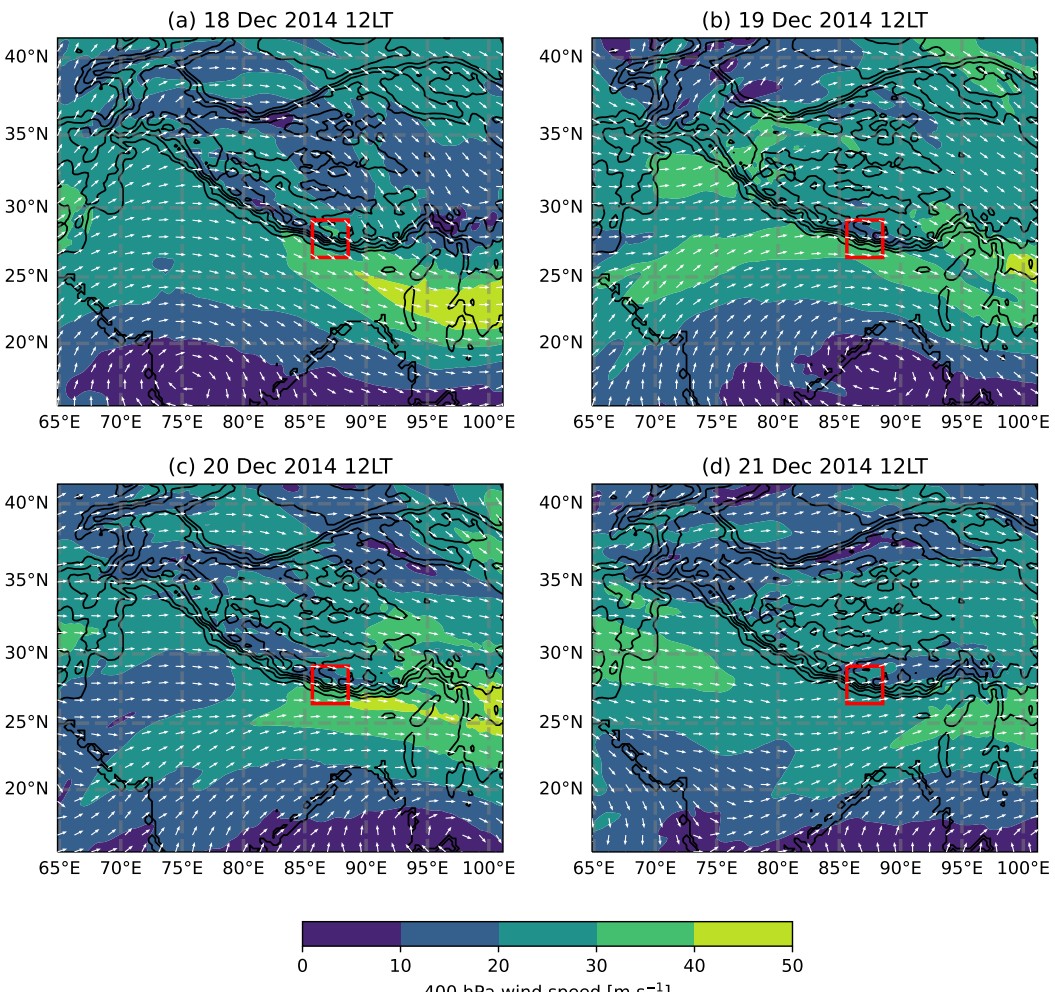

**Figure 4.** 400-hPa wind speed (shading) and direction (white vectors) in the d01 domain at 12 local time (noon) on 18–21 December 2014. Black solid lines show the WRF simulation topography (m.a.s.l) with a contour interval of 1000 m. The red box mark the location of the inner domain d04. 12LT 17 December 2014 is excluded due to the model spin-up time (see Sect. 2.2).

on the sides and at the valley top (Fig. 3d) that shelter the valley atmosphere from the large-scale flow channelling. Up to 25 m s$^{-1}$ near-surface winds are found on the ridges surrounding the Kanchanjunga valley, but within the valley atmosphere the near-surface winds stay below 10 m s$^{-1}$ and flow in the up-valley direction. Due to the potential interruption of the thermally driven winds by the large-scale winds on 18th Dec, and that we want to focus on situations where there is little interaction with the large-scale flow, the analysis mostly concentrates on the 20–21st Dec in the following sections.

The daytime valley winds propagate into the entrances of the two westernmost valleys (Gaurishankar and Khumbu) over the perpendicular topographic barrier from the plain (Figures 2b, 3a-b). Since we focus on the along-valley winds, it is sensible

to extend the valley center lines towards the plain over this barrier instead of following the topography towards the south-east
(Fig. 2b). In this way, the structure of the flow in plain-valley interaction can be studied better (cross-sections are discussed later in Sect. 4.2).

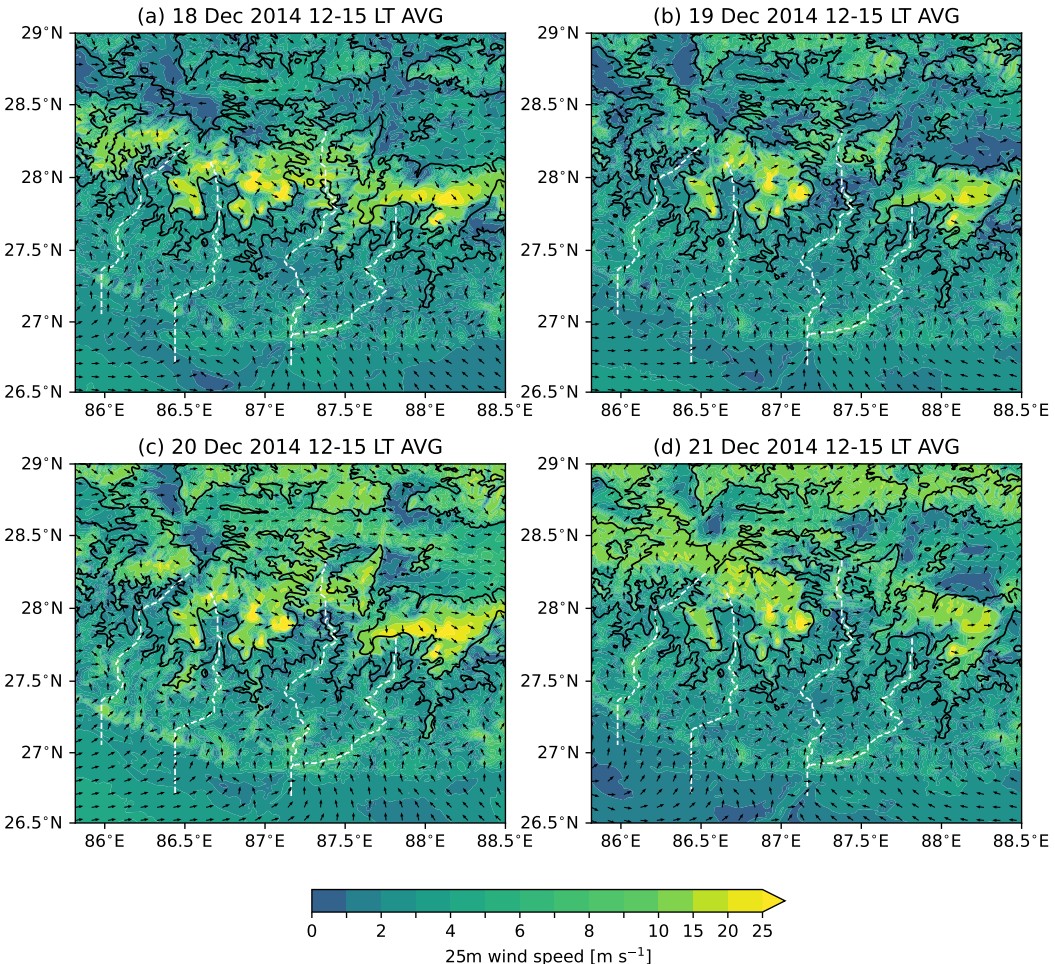

**Figure 5.** Wind speed (shading) and direction (black vectors) on the lowest vertical level (approximately 25 m above surface) in the d04 domain on 18-21 December 2014. The wind is averaged over 12-15 local time for each day of the simulation. Black solid contours show the WRF simulation topography at 3000 and 5000 m.a.s.l. Valley center lines are shown by the white dashed lines. 12–15LT 17 December 2014 is excluded due to the model spin-up time (see Sect. 2.2).

## 4.1 Temporal evolution of the along-valley and cross-valley winds

The along-valley wind component is calculated based on Eq. 4 for four different parts of each valley. These locations are shown by crosses in Figures 2b and 3 and the colors refer to the along-valley wind timeseries in Fig. 6. The along-valley

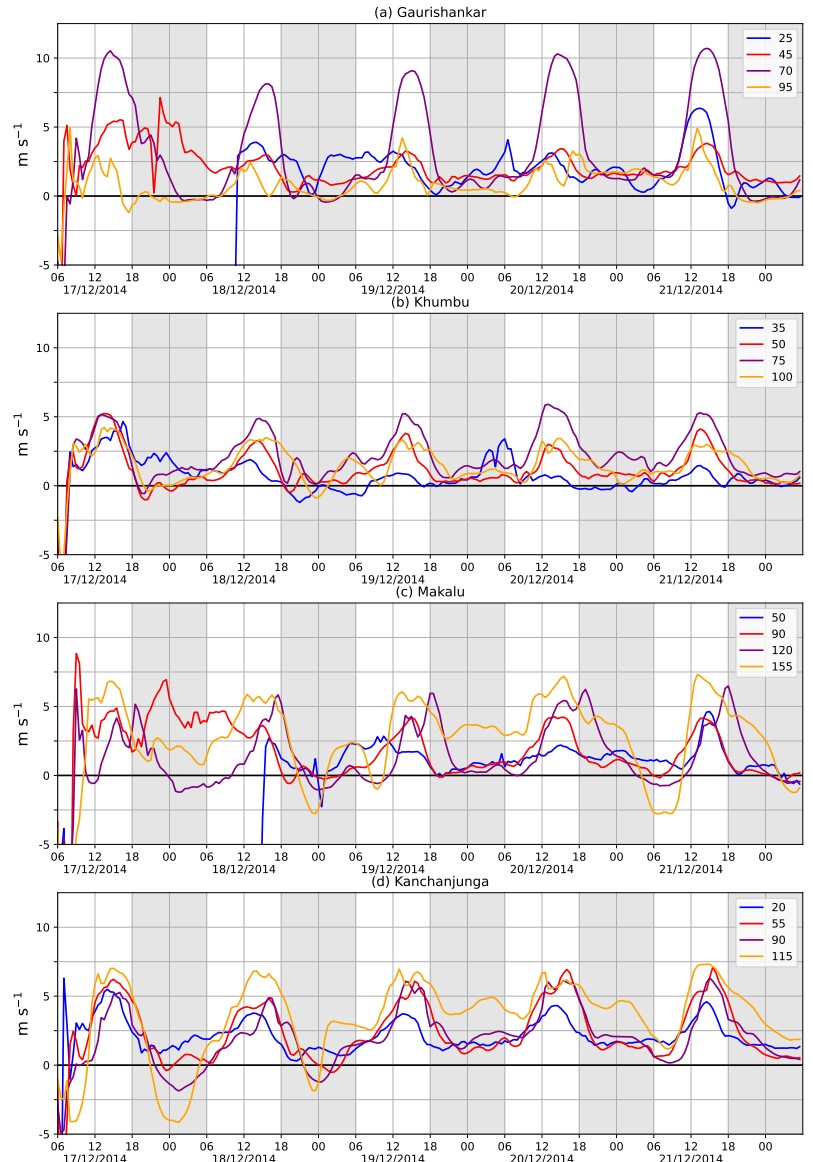

**Figure 6.** Along-valley wind velocity in different parts of each of the four valleys (crosses of the same colors in Figures 2 and 3) on the fourth model level (approximately 300 m above surface). The numbers in the legends refer to the along-valley grid points as plotted on the x-axes of Fig. 3. Positive values of wind velocity refer to up-valley winds. Local time is shown on the x-axes. The data is plotted every 30 minutes. Same analysis with extended y-scale to show the along-valley wind velocities less than -5 m s$^{-1}$ is shown in Supplementary Figure A2.

wind component in Fig. 6 is the average of 5 grid points along the valley center line around the crosses to describe the wind in a larger part of the valley, instead of only at one grid point. The four locations in the valleys represent the valley entrance

(orange), two location in the middle of the valley (purple and red) and the valley top (blue). The description of the along-valley winds is mostly based on the 20-21 Dec when the thermally driven winds were well defined in the valleys, but also the 18-19 Dec are considered in the text. 17 Dec 06–18LT is shown in the timeseries but excluded in the analysis due to being in the model spin-up time (Sect. 2.2).

The along-valley winds have a clear diurnal cycle with well defined daytime up-valley winds and weak or absent nocturnal down-valley winds during 18-21 Dec (Fig. 6). During the study period the up-valley winds start developing around 9 local time (LT), peaks in magnitude around 15 LT and cease around 18 LT in all of the four valleys. The onset and offset of the daytime up-valley winds occurs approximately 2 hours after the sunrise (6:50 LT) and 1 hour after the sunset (17:12 LT), respectively. The along-valley winds at night are weak in all four valleys and generally remain in the up-valley direction (night-time shaded in Fig. 6). During the 17-18 Dec the diurnal cycle is more interrupted than during the 19-21 Dec and the simulated night-time down-valley winds are caused by the channelling of the north-westerlies into the valleys. The high wind speeds caused by the channelling is shown in Supplementary Figure A2.

In the Gaurishankar valley, the daytime up-valley winds have maximum values of 5, 10, 5 and 4–6 m s$^{-1}$ in the four marked locations between the 20-21 Dec, listed from the valley entrance to the valley top (Fig. 6a). During night-time, the winds are around 2 m s$^{-1}$ and directed up-valley in the top half of the valley (grid points 25 and 45 in Fig. 6a) and less than 1 m s$^{-1}$ up or down-valley in the lower half (grid points 70 and 95 in Fig. 6a). In the top of the Gaurishankar valley, the along-valley winds do not show a clear diurnal cycle except on the last day of the simulation. In other parts of the valley, the timing of the diurnal cycle is relatively similar on 18-21 Dec.

In the Khumbu valley, the daytime up-valley winds have maximum values of 3, 5, 4–5 and 2–3 m s$^{-1}$ in the four marked locations between the 20-21 Dec, listed from the valley entrance to the valley top (Fig. 6b). During the simulation period, the wind speed is weakest in the valley top and strongest in the mid-part of the valley (Supplementary Figures A4c,d). During night-time, the along-valley winds remain mostly in the up-valley direction with magnitudes less than 2 m s$^{-1}$ except in the valley top where, on the night between the 18th and 19th, the along-valley wind is weak (< 1 m s$^{-1}$) and in the down-valley direction (Fig. 6). The along-valley variation in the timing of the diurnal cycle is small in the Khumbu valley during the study period. Day-to-day variation in the along-valley winds is small, the along-valley wind speeds vary less than 2 m s$^{-1}$ in each part of the valley and the on-set and off-set of the up-valley winds varies by less than an hour during 18-21 Dec.

In the Makalu valley, the daytime up-valley winds have maximum values of 7–8, 6, 5, 2–5 m s$^{-1}$ in the four marked locations between the 20-21 Dec, listed from the valley entrance to the valley top (Fig. 6c). At the valley entrance the night-time winds vary from 2.5 m s$^{-1}$ down-valley to 5 m s$^{-1}$ up-valley (yellow timeseries, grid point 155 in Fig. 6a). The along-valley winds do not have a clear repetitive diurnal cycle at the valley top except on the last day of the simulation. The timing of the diurnal cycle during 18-21 Dec varies between the different locations along the Makalu valley. In the top-mid-way of the valley (red timeseries, grid point 90 in Fig. 6c) the on-set, peak in magnitude, and off-set of the up-valley winds is similar as in the other valleys during the study period. At the valley entrance, the on-set and peak in magnitude occur at the same time as in the other valleys but the decay of the up-valley winds is slower; on 20th and 21st Dec the along-valley wind component turns down-valley only around 02-03 LT in the night. In the middle of the valley (purple timeseries, grid point 120), the on-set of

the up-valley winds occur at the same time as in the other valleys but there are two maximums, the first around 15LT and the second time around 18LT. The decay of up-valley winds in this location is thus delayed by 3 hours compared to the other valleys.

In the Kanchanjunga valley, the daytime up-valley winds have maximum values of 7–8, 6, 7, 5 m s$^{-1}$ in the four marked locations between the 20-21 Dec, listed from the valley entrance to the valley top (Fig. 6d). During night-time the along-valley winds stay directed up-valley, ranging from 1 to 5 m s$^{-1}$ during 20-21st Dec. The daytime up-valley winds vary only by 2–3 m s$^{-1}$ along the valley and the on-set and off-set times are similar throughout the valley. The amplitude of the diurnal cycle is slightly larger in the middle of the valley than at either the valley entrance or valley top. The up-valley winds extend all the way

to the valley top on each day of the simulation. The decrease in the daytime up-valley winds is slower at the valley entrance compared to the other parts of the valley.

In all of the four valleys, the strongest up-valley winds occur in the entrances of the Makalu and Kanchanjunga valleys and 30 km into the Gaurishankar valley (around the along-valley grid point 70). In the Makalu and Kanchanjunga valleys, these strong valley entrance jets (Supplementary Figure A4e,g) could potentially be explained by the local strong temperature

difference (discussed in detail in Sect. 4.2) between the valley and the plain, which would result as forcing of plain-to-valley winds.

In the 5-day simulation, the peak magnitude in winds in the Gaurishankar valley is located near a narrow (in the cross-valley direction, shown in Supplementary Figure A4a) part of the valley. The strong winds in this location can be explained by the Venturi effect (Whiteman, 2000). The local narrowing of the valley topography accelerates the wind speed through this gap

assuming a constant along-valley massflux on both sides of the narrowing part. The wind speed above this location, 200 m higher than the model level that is shown in Fig. 6, is already reduced by half (not shown). The reduction in the wind speed with height is not as strong in the other locations of the Gaurishankar valley.

As an overview, excluding the parts of the valleys without persistent diurnal cycles in wind speed and the narrow gap channelling, the daytime up-valley winds are 2–3 m s$^{-1}$ stronger in the Makalu and Kanchanjunga valleys compared to in

the Gaurishankar and Khumbu valleys. The Kanchanjunga and Khumbu valleys have the most persistent diurnal cycle in the up-valley winds in this 5-day simulation.

The cross-valley winds are shown in Fig. 7 in a similar manner to the along-valley wind components. However, whereas the along-valley winds were analysed in the center of the valley, the cross-valley winds are analysed separately on both the east and west slopes of the valleys. The cross-valley wind component is considered here as a wind component perpendicular

to the valley center lines (introduced in Sect. 2.3.1) thus perpendicular to the along-valley wind component directly above the valley center line. Positive values for the cross-valley wind component on both slopes refer to up-slope winds. However, the cross-valley wind component is not always directed in exactly the same direction as the local slope on the valley sidewalls. This is because the sidewall slopes are not always perpendicular to the valley center line.

A clear diurnal cycle of the up-slope cross-valley winds is only found in some parts of some valleys. The up-slope winds

occur most often in the middle parts of the valleys, seen as red and purple timeseries in Fig. 7. In the locations with clear

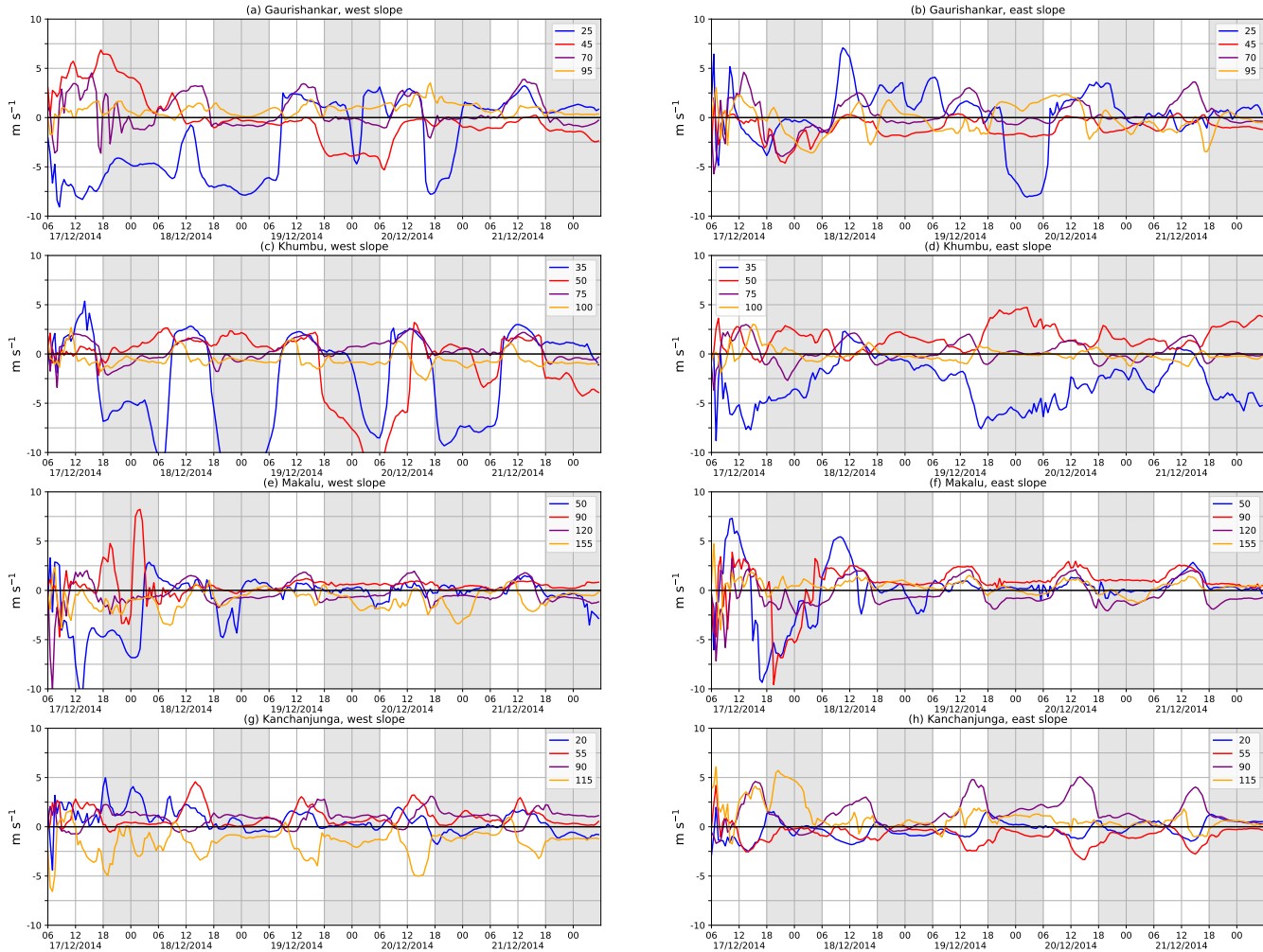

**Figure 7.** Cross-valley wind velocity in different parts of each of the four valleys (see Fig 2b) on the lowest model level (approximately 25 m above surface). Panels in the left column are for grid points located on the western slope of the valleys and panels in the right column for grid points on the eastern slopes. Local time is shown on the x-axes. The cross-valley wind component on the western slope is multiplied by $-1$ to present both slopes in a manner that positive values refer to an up-slope wind. The data is plotted every 30 minutes.

diurnal cycles, the daytime up-slope winds are less than 3 m s$^{-1}$ in magnitude and their vertical extent is less than 300 m (see Supplementary Figures A3–A4 for vertical cross-sections of the wind components across the valleys).

A clear diurnal cycle of cross-valley winds on both the west and east slope at the same point is found only in the middle-parts (red and purple timeseries) of the Makalu and Kanchanjunga valleys during the 19-21st Dec (Fig. 7e-h). These locations refer 395 to along-valley grid points 90 and 120 in Makalu and 55 and 90 in Kanchanjunga. The up-slope winds start to develop right after sunrise at around 7 LT (sun rise 6:50LT), peak in magnitude around noon, and cease around 15LT.

One of the analysed locations, mid-way of Kanchanjunga (red timeseries in Fig. 7d), show signs of a single circulation cell. Such a single circulation cell spans the whole cross-valley direction and can be identified if there are daytime up-slope winds at one slope and down-slope winds at the other slope (Supplementary Figure A3h).

Overall, from the WRF simulation, there is not strong evidence of well-defined thermally driven circulation in the cross-valley direction in the four valleys (Supplementary Figures A3–A4). However, the cross-valley winds shown in Fig. 7 are not the pure slope wind circulation, since the gradient of the slope elevation is not necessarily aligned perpendicularly to the valley center line. Also, the lowest three model levels at heights 25m, 90m and 190 m may not capture the thermally driven cross-valley circulation, due to the shallower nature of slope winds and the fact that nighttime inversions may be below the

lowest model level. Up to 10 m s$^{-1}$ wind speeds are simulated at the surface at higher elevations (i.e in the top parts of the Gaurishankar and Khumbu valleys, Fig. 7a-d). Some of the chosen grid points for cross-valley analysis may be located outside the valley atmosphere and thus may show strong near-surface winds due to impact of large-scale winds.

## 4.2    Vertical structure of the along-valley winds and potential temperature

The vertical structure of the along-valley wind component and potential temperature above the valley center lines are plotted in

Figures 8 and 9 on 20th Dec at 9LT and 15LT, respectively. The wind component parallel to the valley center lines is plotted on each model level. The large variability in the along-valley direction in the along-valley wind speed above the up-valley wind layers arises because winds above the valley atmosphere are stronger, and their direction influenced by the large-scale flow and not the local topography. Since this study concentrates on the daytime up-valley wind layer, the colour map for the wind shading is not optimal for the wind speeds exceeding 10 m s$^{-1}$. The cross-sections show the whole length of the valley center

lines (yellow lines in Fig. 2b), which means that the far right-hand side of all the panels in Figures 8–9 are the wind component and potential temperature above the plain.

In the morning (20th Dec 09LT - 2 hours and 10 minutes after sunrise), the valley atmospheres are mainly characterised with weak up-valley winds (Fig. 8). The wind speed is mostly less than 2 m s$^{-1}$ in the valleys Gaurishankar and Khumbu and the flow depth above the valley center line is 500-1000 m. In the Makalu and Kanchanjunga valleys the wind speed is up to

4 m s$^{-1}$ and the flow depth is 1000-1500 m above the valley center lines. The up-valley winds exceeding 3 m s$^{-1}$ cover half of the along-valley distance in Kanchanjunga where as in Makalu they are found only at the valley entrance. Some parts of the valleys have weak (< 2 m s$^{-1}$) near-surface down-valley winds which flow in shallow layers of less than 200 m. These weak down-valley winds are found in all of the valleys but they only cover a minor share of the along-valley distance and the up-valley winds are the dominant feature of the valley atmospheres even in the morning.

In the morning (20th Dec 09LT), the potential temperature difference between the valley atmosphere and the plain is small in the layer where the up-valley winds flow (Fig. 8). This is seen as mostly horizontal isentropes between the valley atmosphere and the air above the plain i.e. the lowest 1500 m from the ground. In all of the valleys, and especially in their sloped parts, the isentropes turn towards the surface in a layer of 100-200 m, meaning that the valley floor is heated even in the morning at 09LT. In the Kanchanjunga valley (Fig. 8d), the isentropes tilt towards the surface already in the morning all the way in the

valley in a layer of about 300 m deep, meaning the valley atmosphere is warmer than the air above the plain. This is consistent with the stronger up-valley winds at this time in Kanchanjunga compared to the other valleys.

In the afternoon (20th Dec 15LT), the up-valley winds flow in the valley atmosphere all the way to the top of the valley (blue crosses in Fig. 9) in all of the four valleys. The up-valley wind speeds vary from 2 to 10 m s$^{-1}$ in the afternoon (described in details in Sect. 4.1). The up-valley wind maxima is found around the same height in all of the valleys which is around 200–300

m above the valley center line. In the lower parts (from the yellow crosses to purple crosses) of the valleys, the up-valley winds flow in a deeper layer compared to the upper parts (from red crosses to blue crosses). Figure 9 shows that in the Kanchanjunga valley the up-valley wind speed and depth are the most consistent along the valley. The valley entrance jets are seen around the grid points 140-160 in the Makalu valley and 100-120 in the Kanchanjunga valley (Supplementary Figure A3e, g). The narrow gap flow in the Gaurishankar valley (discussed in Sect. 4.1), around the purple cross (grid point 70, Supplementary

Figure A3a), is seen as a relatively small area of up-valley winds exceeding 10 m s$^{-1}$ (also in Supplementary Figure A4a). In the top of the Gaurishankar and Khumbu valleys, the depth of the up-valley wind layer decreases to a few hundred meters yet still extends all the way to the top of the valleys (Supplementary Figure A4b, d). In the afternoon, all of the valley atmospheres are 3–5 K warmer in the mid-way of the valley (purple crosses) compared to the same altitude above the plain (Fig. 9).

The plain-to-valley winds flow over the 1000 m high barrier at the entrances of the Gaurishankar and Khumbu valleys in

a shallow layer of less than 500 m (grid points 110-130 in Gaurishankar and 120-140 in Khumbu). The plain-to-valley winds basically stop between lee side of the barrier and the valley entrance (around grid point 110). Both in the morning (Fig. 8a-b), and in the afternoon (Fig. 9a-b), the wind component towards the valleys decreases to 1 m s$^{-1}$ for the whole depth of the plain-to-valley wind layer. Along the lee side of the barrier, between the plain and the Gaurishankar and Khumbu valleys, the isentropes descend and are parallel to the slope (Figures 9a–b ). This results in a warming of 5 K (Gaurishankar) and 3 K

(Khumbu) in the lee of the barrier compared to the base of the barrier over the plain.

Figure 10 shows the potential temperature at 15LT minus the potential temperature at 09LT as a vertical cross-section above the valley center lines on Dec 20th and 21st. The up to 1.5 K warming between 09 and 15 LT on Dec 20th and cooling between 09 and 15 LT on Dec 21st in the higher altitudes (> 2000 m above the surface) occurs most likely due to large-scale thermal advection. The valley atmospheres warm 4–6 K between 09 and 15LT on Dec 20th. The surface-based warming extends to

depths of less than 1000 m in the valleys Gaurishankar and Khumbu and to depths of less than 1500 m in the valleys Makalu and Kanchanjunga. The strongest warming between 09LT and 15LT is found in the near-surface layer i.e. the closest 100-200 m to the surface. The warmed layer in the valley atmospheres decrease in magnitude and depth towards the top of the valleys. Where the valley floors start to incline strongly up towards the tops of the valleys is approximately where the warmed layer starts to get shallower. Unlike in the other three valleys, the valley floor in the Kanchanjunga valley only starts to incline

strongly close to the top of the valley and the near-surface warming reaches further into the Kanchanjunga valley compared to the other three valleys.

The qualitative difference between the spatial pattern of the thermal structure of the valley atmospheres on Dec 20th and Dec 21st is small (Figures 10 left column compared to 10 right column). The strongest warming is located around same parts of the valleys on both days. Considering that the daytime warming is also affected by the large-scale weather particularly above the

valley atmosphere, the vertical extent of the strongest warming in the valley atmospheres is similar between the two days. The warming is 1-2 K stronger on Dec 21st than on the Dec 20th which is consistent with the up-valley wind speeds being stronger on Dec 21st compared to Dec 20th (Fig. 6).

Figure 11 shows the deviation of potential temperature at each along-valley grid point (y-axis) on three model levels from the two day average of potential temperature during Dec 20-21st. Here we define the amplitude of the diurnal cycle to be the
difference between the maximum and minimum value of $\theta - \bar{\theta}$ in Fig. 11.

Figures 10 and 11 show that the diurnal cycle in potential temperature reaches higher up in the valley atmosphere compared to over the plain. At the lowest model level (approximately 25 m above surface, Figures 11a-d), the diurnal cycle has a larger amplitude over the plain (up to 8 K) than in the valleys (4–6 K) but the amplitude decreases rapidly with height over the plain. At the height of approximately 450 m above surface (model level 4, Figures 11e-h), the amplitude of the diurnal cycle is less
than 1 K above the plain, where as the amplitude reaches up to 3 K in the Gaurishankar and Khumbu valleys and up to 5 K in the Makalu and Kanchanjunga valleys. The vertical extent of the warming is more evident in the cross-sections shown in Fig. 10, as the layer in which the surface-based warming exceeds 2 K between 09LT and 15LT is hundreds of meters deeper than above the valley.

In addition to the qualitative overview of the vertical structure of the along-valley winds and potential temperature field, a
more detailed analysis of how the depth of the daytime up-valley wind layer and the depth of the warmed layer relate to each other is now considered. The depth of the up-valley wind layer is defined based on the model level at which the clear diurnal cycle vanishes in the along-valley wind timeseries (not shown but one model level shown in Fig. 6). Similarly, the vertical extent of the amplified diurnal cycle in potential temperature is based on the model level at which the daily range in potential temperature is the same as over the plain (not shown but three model levels shown in Fig. 11).

The depth of the up-valley wind layer and the vertical extent of the amplified diurnal cycle in potential temperature were defined based on the model level at which the clear diurnal cycle vanishes compared to the plain in the along-valley wind timeseries (not shown but one model level shown in Fig. 6) and in the amplitude of the diurnal cycle of potential temperature (not shown but three model levels shown in Fig. 11), respectively.

In the Gaurishankar and Khumbu valleys, the diurnal cycle in potential temperature and the up-valley winds are found in a
layer 600–1200 m deep above the valley center lines, whereas in Makalu and Kanchanjunga the corresponding depth is 1000-1500 m. In all of the valleys during the 5–day simulation, both the vertical extent of the diurnal cycle in potential temperature and the layer of up-valley winds is deepest in the lower parts of the valleys (yellow and purple crosses). The up-valley wind layer is deepest in the portion of the Makalu (grid points 120-160) and Kanchanjunga (grid points 70-120) valleys with flat valley floor, where the flow depth reach up to 1500 m, and shallowest in the the top of the valleys Gaurishankar (grid points
20–40) and Khumbu (grid points 30-50), where the flow depth is less than 600 m.

The vertical extent of the amplified diurnal cycle in temperature seems to correlate with the up-valley wind layer depth in the valleys; where the amplified diurnal cycle in temperature reaches higher, also the daytime up-valley winds reach higher. When the temperature in the valley atmosphere rises more compared to the air above the plain during daytime, from the hydrostatic law and ideal gas law combined one obtains the qualitative result that pressure at the same height must be smaller in the valley

than over the plain. Thus, if the winds are driven by the pressure gradient force, the depth of the heated layer (i.e. in the valley atmosphere) would correlate with the depth of the daytime up-valley wind layer if pressure is horizontally homogeneous above the heated layer.

## 5 Discussion

We now attempt to understand the differences in the daytime up-valley winds during this 5-day period between the four
valleys, and along the individual valleys, and relate these differences to the differences in the valley topographies. To do this, we relate our results to previous studies which have used highly controlled idealised simulations to quantify the impact of valley geometry on valley winds. We also assume that since all four valleys are under similar large-scale forcing that the differences in the along-valley winds are mainly due to differences in the valley topographies.

Two features that differ significantly in the valley topographies are the along-valley variation of the valley floor inclination
and the topography of the valley entrances. As summarised in Sect. 3, the two westernmost valleys, Gaurishankar and Khumbu, have inclined valley floors throughout the length of the valley (1–2 degrees in the lower part of the valleys and 2–5 degrees in the upper part) and an 1 km high perpendicular barrier between the valley entrance and the plain. Makalu and Kanchanjunga, on the other hand, have a 40 km long almost flat portion close to the valley entrance and no barrier.

During the 5–day study period, the daytime up-valley winds are both weaker and flow in a shallower layer in the parts of
the valleys where the valley floor has a steep inclination (up to 5 degrees). Wagner et al. (2015) studied the influence of valley geometry (floor inclination, width, depth, valley cross-section narrowing) on thermally driven flows using idealised numerical simulations. When they compared two straight valleys, one with a flat valley floor and the other with an inclination of 0.86 degrees in the valley floor, they found that the daytime up-valley wind speed increased by a factor of 3.0 in the valley with inclined floor. Wagner et al. (2015) suggested that the increase in wind speed was due to both the reduction of the valley
volume by 50% and to additional buoyancy forcing from the slope wind effect. Our finding, that the strongest up-valley winds occur in the flatter parts of the valleys, appears to contradict the result of Wagner et al. (2015). However, in the four Himalayan valleys, the ridge height also increases along the valley whereas in the idealised valleys Wagner et al. (2015) studied the ridge height was constant along the valley. Therefore, in the Himalayan valleys, the steeply inclined valley floors do not necessarily lead to a reduced valley volume and enhanced topographic amplification factor along the valleys. In addition, the valley floors
slope much more steeply in the Himalayan valleys compared to the valleys in Wagner et al. (2015). This may mean that the dominate driving mechanism of the up-valley winds differs in our simulations compared to in the simulations of Wagner et al. (2015). Specifically, in steeply inclined valleys, the buoyancy mechanism that drives up-slope winds in classical mountain wind theories (Whiteman, 2000), may become more dominate than the valley-wind mechanism. The buoyancy mechanism drives shallower and weaker winds, such as the typical cross-valley slope winds, compared to the valley wind mechanisms. Shifting
the dominant driving mechanism from the valley volume effect to the buoyancy mechanism, instead of combining their forcing, would explain the shallower and weaker up-valley winds in the steeply inclined parts of the four Himalayan valleys.

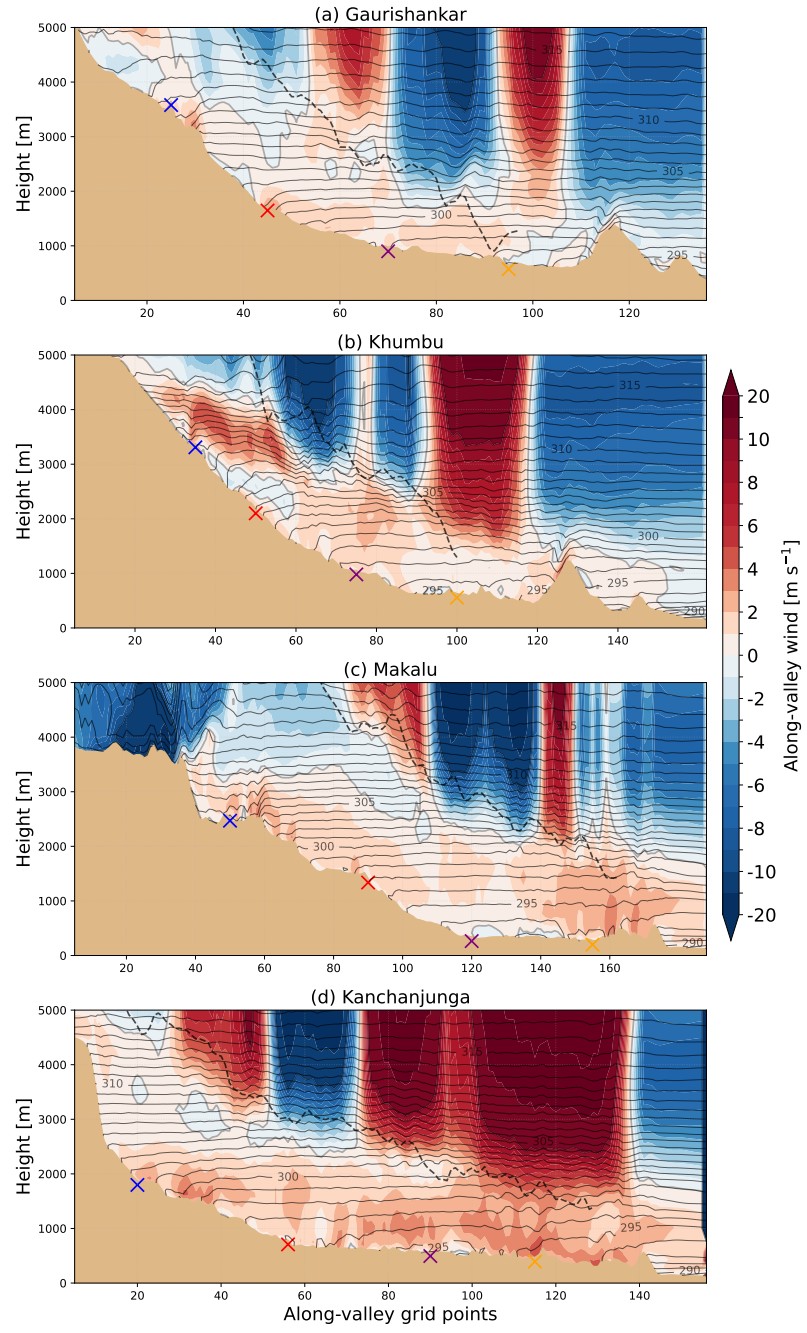

**Figure 8.** Vertical cross-section of potential temperature (grey contours) and the along-valley wind component (shading) above the valley center lines (yellow lines in Fig 2) on 20th Dec at 09 local time. Potential temperature is plotted with a contour interval of 1 K. Positive values for the along-valley wind refer to up-valley winds. The mean of the two ridge heights is shown by the black dashed lines.

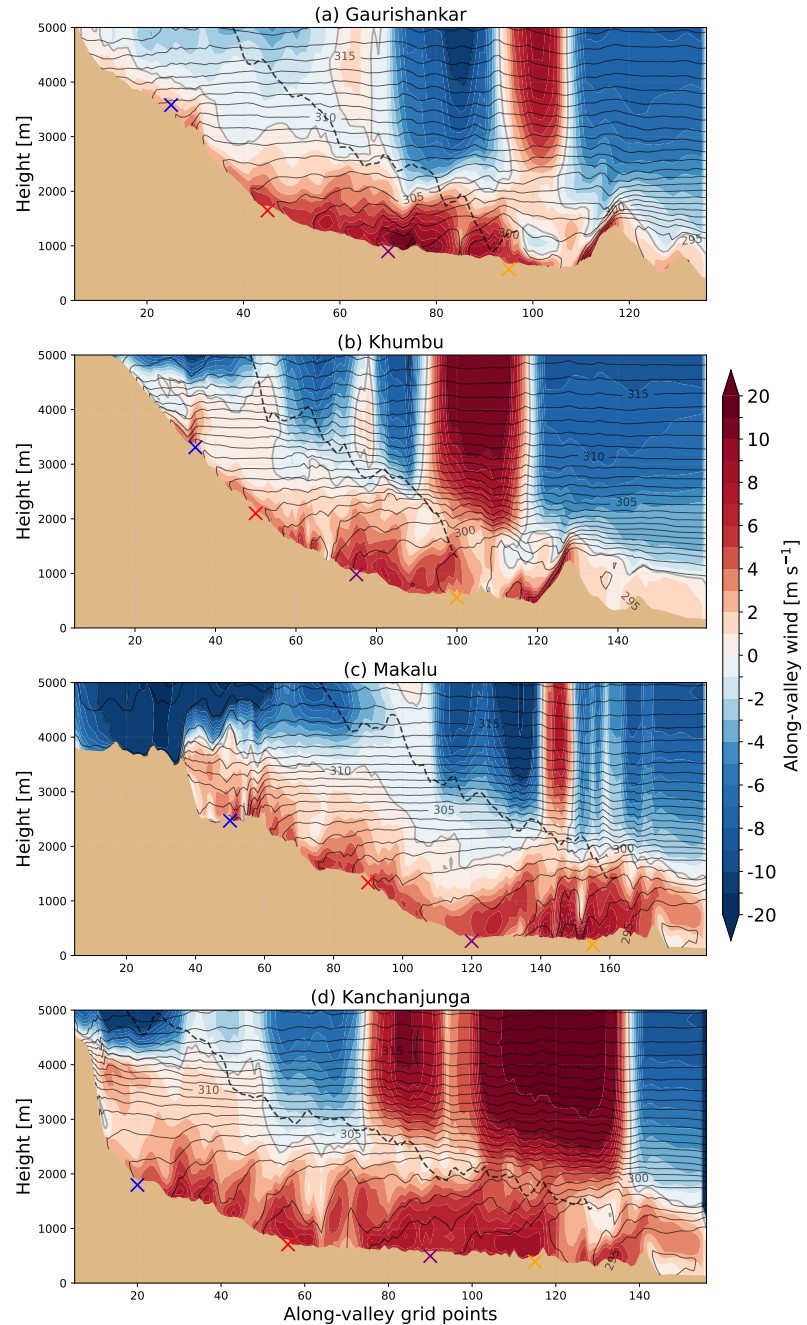

**Figure 9.** Vertical cross-section of potential temperature (grey contours) and the along-valley wind component (shading) above the valley center lines (yellow lines in Fig 2) on 20th Dec at 15 local time. Potential temperature is plotted with a contour interval of 1 K. Positive values for the along-valley wind refer to up-valley winds. The mean of the two ridge heights is shown by the black dashed lines.

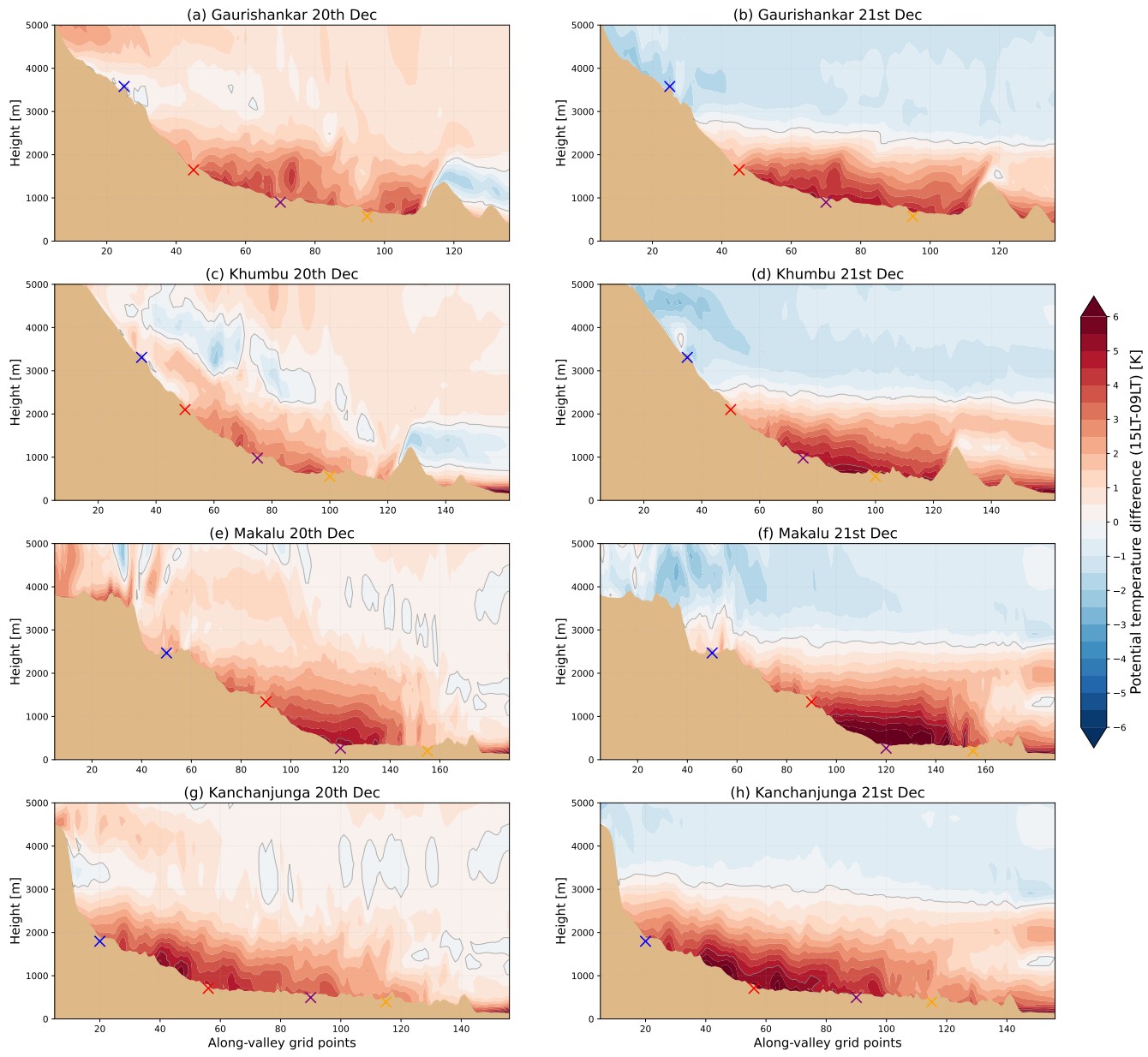

**Figure 10.** Vertical cross-section of potential temperature change above the valley center lines (yellow lines in Fig 2b) between 09LT and 15LT on Dec 20th (left column) and 21st (right column). Positive values indicate higher potential temperatures at 15 LT than at 9 LT.

Instead of open valley entrances, the valleys Gaurishankar and Khumbu have the 1 km high barrier between the valleys and the plain. As discussed in Sect. 4.2, the daytime plain-valley winds cross this barrier in a shallow layer of a few hundred meters. In contrast, in the Makalu and Kanchanjunga valleys, the plain-to-valley winds flow in layers as deep as 1000–1500 m. The

up-valley winds basically stop on the northern side of the barrier close to the valley entrances of Gaurishankar and Khumbu. The up-valley wind propagation into the valley could be significantly interrupted by the flow forced to flow over the 1 km high barrier. The flow characteristics over this barrier are similar to what Stull (1988) describes the down-slope wind storms associated with hydraulic jump. The strong and shallow flow over the barrier is followed by weaker horizontal winds lee-side after the barrier.

Bianchi et al. (2021) suggested that the daytime up-valley winds in the Khumbu valley would transport aerosol precursors from the bottom of the valley to the free troposphere. During this transport, these gases are oxidised and therefore able to form new particles and influence the climate once they are in the free troposphere. They combined in-situ aerosol observations and numerical model simulations with the high resolution WRF-model and the Langrangian dispersion model FLEXPART. They propose that other valleys on the southern slope of the Himalaya would also act as sources of free tropospheric aerosol. We show that regarding the daytime up-valley winds, the Khumbu valley is not an exception compared to the other major valleys in this region. Based on the along-valley winds, aerosol and its precursors could be ventilated into the free troposphere from the other three valleys as well.

## 6 Conclusions

The local valley winds in the Nepal Himalayas during 17-21 December 2014 were studied using a high-resolution WRF simulation. The horizontal grid spacing in the inner domain is 1 km and the model is run with 61 vertical levels. A 12 hour spin-up period is applied, hence the first 12 hours of the 5–day simulation is excluded from the analysis.

Four major valleys are present in the inner-most model domain and the characteristics of the along-valley winds that develop in each of these valleys during this 5-day period were analysed and compared to each other. These Himalayan valleys have very different topographies compared to the much more extensively studied valleys in the European Alps and Rocky Mountains. Specifically, the floor of the Himalayan valleys studied here are steeply inclined and rises from less than 500 m.a.s.l to 4000-5000 m.a.s.l in 100 km in the along-valley distance.

The simulation is evaluated using meteorological observations from three automatic weather stations in the Khumbu valley. Overall, the model manages to simulate the diurnal cycle of the winds that was evident in the observations. The along-valley wind characteristics in the Khumbu valley were found to be similar to what previous research has shown both in observational (Inoue, 1976; Ueno and Kayastha, 2001; Bollasina et al., 2002; Ueno et al., 2008; Bonasoni et al., 2010; Shea et al., 2015) and model-based studies (Potter et al., 2018, 2021).

Daytime up-valley winds are found in all of the four valleys during the simulated 5-day period. The night-time along-valley winds are weak in strength and flow mostly in the up-valley direction. During large-scale northerlies and north-westerlies (Dec 17–18th), the daily cycle of along-valley winds is interrupted more compared to the days with large-scale westerlies (Dec 20th-21st) especially in the tops of the valleys Gaurishankar, Khumbu and Makalu. The Kanchanjunga valley is an exception in the valleys, as the daytime up-valley winds reach the top of the valley even during the days with large-scale north-westerlies.

The night-time down-valley winds are found more during the large-scale northerlies, which is most likely due to channelling of above-valley winds into the valley atmosphere.

During the simulated 5 days the daytime up-valley winds vary between the valleys and their parts both in strength and flow depth. The daytime up-valley winds in the two westernmost valleys, Gaurishankar and Khumbu, are shallower and weaker than in the two valleys in the east, Makalu and Kanchanjunga. These two group of valleys can be separated by their topography characteristics: the valleys in the west have a continuous inclination in the valley floor and there is an 1 km high perpendicular mountain barrier between the valley entrance and the plain. The two valleys in the east have a 40 km (Makalu) and 60 km (Kanchanjunga) portion with flat valley floor from the open valley entrance into the valley.

Steep inclination in the valley floor (2–5 degrees) is associated with weaker and shallower up-valley winds compared to locations with nearly flat valley floor (<1 degrees inclination). The perpendicular barrier at the valley entrance potentially interrupt the daytime plain-to-valley wind propagation which is seen as weaker daytime up-valley winds at the valley entrance and potentially leading to weaker up-valley winds further up in the valley.

These results, obtained from a relatively short but high spatial and temporal resolution WRF simulation, are likely representative of the month of December and the post-Monsoon season as both the local near-surface winds and 400-hPa winds are similar to climatology. However, these results may not be valid in other seasons when cloud cover and the large-scale flow differ. Future investigations could consider other seasons and, if computational resources permit, longer simulations could be performed and analysed.

*Acknowledgements.* This work was supported by the European Research Council with the project CHAPAs no. 850614. We acknowledge Franco Salerno and Nicolas Guyennon from Water Research Institute (Brugherio, Italy) for providing the meteorological observations in Khumbu valley. Observation data available at *http://geonetwork.evk2cnr.org*. We thank Emily Potter and one anonymous referee for their comments which helped improve this manuscript.

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

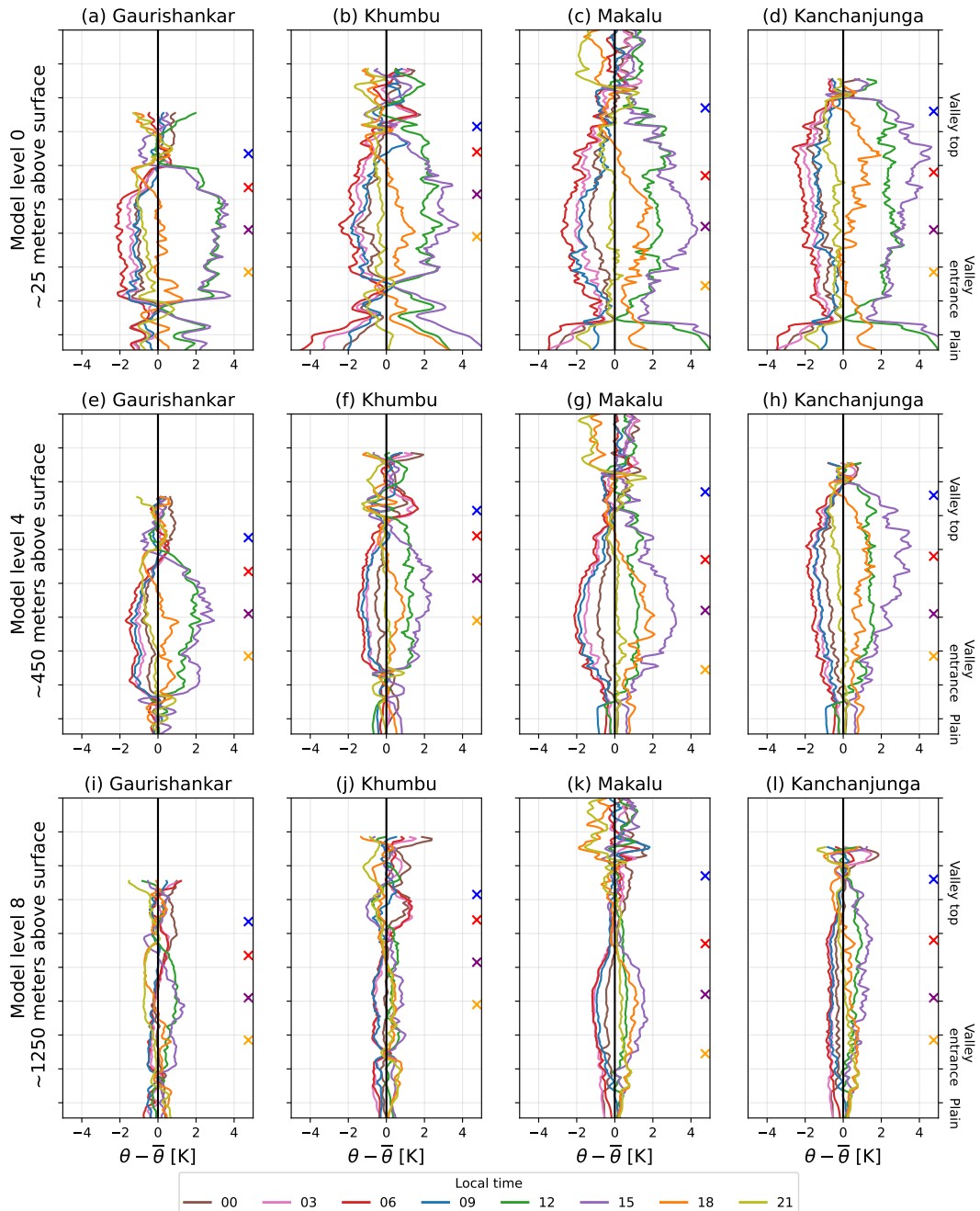

**Figure 11.** Diurnal variations in potential temperature in the valleys at the model levels 0 (a-d), 4 (e-h) and 8 (i-l) during the last two days of the simulation (20-21st Dec 2014). The model levels refer to heights of **a-d)** 25 m, **e-h)** 450 m and **i-l)** 1250 m above the surface. x-axes show the deviation from the two day mean potential temperature of that model level in each grid point located on the valley center line. y-axes are the grid points at the valley center and the crosses on the right hand side of each figure denote the location of the grid points marked on same colors in Figures 2b and 3.