# Peer review of "Daytime along-valley winds in the Himalayas as simulated by the WRF-model"

_EGUsphere, 2022_

## Referee Comment (RC1)

This manuscript presents comparisons of the valley circulations from four neighbouring valleys in the Nepal Himalaya, using the Weather Research and Forecasting model. The wind and potential temperature are compared over five days in the dry season, and variations between the valleys are attributed to topographical differences. In addition, the effect of large-scale weather on valley circulations is considered, and the authors find that northerly winds cause greater disruption to the valley circulations than westerly winds. I found the discussion of up-valley wind speed increases due to narrowing of valley topography particularly interesting, along with the hypothesised switch in drivers from the valley volume effect to increases in buoyancy, in valleys with steeper valley floors.

Overall, the manuscript is well presented, and the results are clearly described. The valley circulations are thoroughly analysed, and the authors have carefully considered both along-valley and cross-valley wind changes, as well as the temperature profiles.

Scientific comments

1. My main comment is that I think the manuscript would benefit from a longer model run. I appreciate that the authors are running at high resolution over a relatively large domain, but I think that a model run longer than five days is needed to make climatological conclusions about the differences between the valleys, or the differences with different large-scale weather patterns.
   If it is computationally impractical for the model to be run for a few weeks or for all of December, perhaps case studies could be picked to analyse a few instances of large-scale northerly winds and westerly winds, for example. Alternatively, strong justification is needed as to why these few days are representative of the broader climatology, however please also see point 2 below about spinup time, which may cut the run even shorter.
2. It's not clear whether the model was run with any spinup time. In general, meteorological features take at least a number of hours to spinup in high-resolution regional climate models (e.g. Bonekamp et al., (2018), Jankov et al., (2007), Román-Cascón et al., (2015)), and this period is normally discarded before analysis. If the model was run without spinup, I suggest this may be the cause of the atypical wind speeds in the first twelve to twenty-four hours in figures 5 and 6.
3. For the comparison in figure A1, I would recommend adjusting the model wind height of 10 m to the observation wind height of 5 m, using a logarithmic profile (see Whiteman (2000)).
4. Lines 272-280 and onwards: as discussed in point 1, comments like 'typically up-valley winds have speeds of….' Or discussion of the variation in timings of the diurnal cycle require a longer time period to assess, especially where only two or three days from the run is being discussed.
5. Figure 5: what are the initial very strong negative wind velocities I the first two days at the top of the Gaurishankar and Makalu valleys? If they are too strong to display easily on the graphs, perhaps mention the minimum value in the figure caption? I'd also recommend adjusting the scale in Gaurishankar to show the top peak.
6. Line 297-302: is the domain big enough to check whether there is a local strong temperature difference between the valley and the plain? Or is this evident from figure 9?
7. If there's room, or perhaps as an appendix, it would be interesting to see some vertical cross sections across the valley, e.g. to illustrate the hypothesised single circulation cell in Kanchanjuga mentioned on line 327.

Typographical comments

8. Table 1 and Figure A1: recommend keeping the name for the pyramid station the same.
9. Figure 1: include the coloured crosses in the figure caption
10. Figure 4: wind vectors in valleys are difficult to see. I recommend using the same colour scheme as in figure 3, and making the topography lines much thinner or sparser.
11. Section 4.1 it would be worth adding more of the along-valley grid point numbers into the text, e.g. line 281 'At the valley entrance…' would be easier to understand as 'at the valley entrance (155 in fig 5 c)…' Or mention the line colour as you have done at other points.
12. Figures 5 and 6: 'wind speed' should be 'wind velocity' as speed cannot be negative.
13. Figure A1: adjust scales to the full model output can be seen (unless this is due to the model still stabilising during a spin-up period, in which case discard this data). Y-axis needs a label (local time or UTC?).

**References**

Bonekamp, P.N.J., Collier, E. and Immerzeel, W.W., 2018. The impact of spatial resolution, land use, and spinup time on resolving spatial precipitation patterns in the Himalayas. Journal of Hydrometeorology, 19(10), pp.1565-1581.

Jankov, I., Gallus Jr, W.A., Segal, M. and Koch, S.E., 2007. Influence of initial conditions on the WRF–ARW model QPF response to physical parameterization changes. Weather and Forecasting, 22(3), pp.501-519.

Román-Cascón, C., Steeneveld, G.J., Yagüe, C., Sastre, M., Arrillaga, J.A. and Maqueda, G., 2016. Forecasting radiation fog at climatologically contrasting sites: evaluation of statistical methods and WRF. Quarterly Journal of the Royal Meteorological Society, 142(695), pp.1048-1063.

Whiteman, C.D., 2000. Mountain meteorology: fundamentals and applications. Oxford University Press.

---

## Author Response (AR1)

**Response to Referees – Daytime along-valley winds in the Himalayas as simulated by the WRF–model**

Johannes Mikkola, Victoria Sinclair, Marja Bister, Federico Bianchi

August 2022

We thank the reviewers for their constructive comments on our submitted manuscript. We have copied the comments of the reviewers in black here and include our response to each individual comment in blue. We have added a new figure as Figure 1 in the updated manuscript, meaning that the numbering of the other figures has changed. In this response, we refer to the NEW figure numbers.

**RC1 - Emily Potter**

This manuscript presents comparisons of the valley circulations from four neighbouring valleys in the Nepal Himalaya, using the Weather Research and Forecasting model. The wind and potential temperature are compared over five days in the dry season, and variations between the valleys are attributed to topographical differences. In addition, the effect of large-scale weather on valley circulations is considered, and the authors find that northerly winds cause greater disruption to the valley circulations than westerly winds. I found the discussion of up-valley wind speed increases due to narrowing of valley topography particularly interesting, along with the hypothesised switch in drivers from the valley volume effect to increases in buoyancy, in valleys with steeper valley floors.

Overall, the manuscript is well presented, and the results are clearly described. The valley circulations are thoroughly analysed, and the authors have carefully considered both along-valley and cross-valley wind changes, as well as the temperature profiles.

**Scientific comments**

1. My main comment is that I think the manuscript would benefit from a longer model run. I appreciate that the authors are running at high resolution over a relatively large domain, but I think that a model run longer than five days is needed to make climatological conclusions about the differences between the valleys, or the differences with different large-scale weather patterns.

If it is computationally impractical for the model to be run for a few weeks or for all of December, perhaps case studies could be picked to analyse a few instances of large-scale northerly winds and westerly winds, for example. Alternatively, strong justification is needed as to why these few days are representative of the broader climatology, however please also see point 2 below about spinup time, which may cut the run even shorter. Firstly, we would like note that we did not aim to draw climatological conclusions based on this short simulation. We have revised the abstract and the introduction to set the aim of the study more clear. Due to computational resources, and also data storage limitations, we could not run the simulations and save the high temporal resolution output for much longer. However, we acknowledge that justification is needed for our choice of days. We now attempt to provide strong justification for why these few days are representative of the broader post-Monsoon time. This is presented in the new section 2.1 which we have added to the manuscript

2. It's not clear whether the model was run with any spinup time. In general, meteorological features take at least a number of hours to spinup in high-resolution regional climate models (e.g. Bonekamp et al., (2018), Jankov et al., (2007), Román-Cascón et al., (2015)), and this period is normally discarded before analysis. If the model was run without spinup, I suggest this may be the cause of the atypical wind speeds in the first twelve to twenty-four hours in figures 5 and 6. We agree that the first 12 hours of the simulation should be not be considered for spin-up reasons. Text about this, including reference to the study by Bonekamp et al., (2018), has now been added to Section 2.2 (Section 2.1 in the first submission). Figures 4 and 5 (Figures 3 and 4 in the first submission) have been updated by excluding the 17 Dec 2014 due to the spin-up period.

3. For the comparison in figure A1, I would recommend adjusting the model wind height of 10 m to the observation wind height of 5 m, using a logarithmic profile (see Whiteman (2000)). Thank you for the suggestion. We have now done this assuming a neutral stratification and revised the comparison to observations with the adjusted height (Table 2 and Supplementary Figure A1). This adjustment of the modelled wind speeds improved the mean absolute errors from 0.6-3.8 m s$^{-1}$ to 0.4-2.5 m s$^{-1}$. In particular, the adjusted modelled daytime maximum wind speeds are closer to the observed than the unadjusted 10-m wind speeds.

4. Lines 272-280 and onwards: as discussed in point 1, comments like 'typically up-valley winds have speeds of....' Or discussion of the variation in timings of the diurnal cycle require a longer time period to assess, especially where only two or three days from the run is being discussed. In this we meant typically in the days we considered, not typical of a long term climate. We have carefully revised the language in the manuscript to be clear that we do not aim to draw climatological conclusions.

5. Figure 5: what are the initial very strong negative wind velocities I the first two days at the top of the Gaurishankar and Makalu valleys? If they are too strong to display easily on the graphs, perhaps mention the minimum value in the figure caption? I'd also recommend adjusting the scale in Gaurishankar to show the top peak. The strong near-surface winds

are seen also in Figure 5 (Figure 4 in the first submission), on top of the crest the wind velocities are above 15 m s$^{-1}$. The winds at around 190 meter height (Fig 6 in the updated manuscript) have minimum values of -25 m s$^{-1}$ (see new Supplementary Figure A2 added to the updated manuscript). The y-scale in Figure 6 (Figure 5 in the first submission) has now been adjusted to show the daytime maximum values.

6. Line 297-302: is the domain big enough to check whether there is a local strong temperature difference between the valley and the plain? Or is this evident from figure 9? The temperature difference between the plain and the valley is evident in Figure 10 (Figure 9 in the first submission). We revised this sentence to refer to section 4.2 where we discuss the temperature difference in more detail.

7. If there's room, or perhaps as an appendix, it would be interesting to see some vertical cross sections across the valley, e.g. to illustrate the hypothesised single circulation cell in Kanchanjuga mentioned on line 327. Cross-valley cross-sections are now added as a new Supplementary Figures A3–A4 in the updated manuscript. Figure A3h shows this particular part of Kanchanjunga referred to in this comment and we now refer to this new Figure from the main text.

**Typographical comments**

8. Table 1 and Figure A1: recommend keeping the name for the pyramid station the same. Thanks for pointing that out. In the revised manuscript, only "NCO-P" is now used when referring to this station.

9. Figure 1: include the coloured crosses in the figure caption
The caption of the Figure (Figure 2 in the updated manuscript) has now been revised.

10. Figure 4: wind vectors in valleys are difficult to see. I recommend using the same colour scheme as in figure 3, and making the topography lines much thinner or sparser. Thanks for the suggestion. The same color scheme as used in the original Figure 3 (Figure 4 in the updated manuscript) is now used in this figure. The number of topography contours has also been reduced to two (3000 and 5000 masl) instead of contours every 1000m.

11. Section 4.1 it would be worth adding more of the along-valley grid point numbers into the text, e.g. line 281 'At the valley entrance...' would be easier to understand as 'at the valley entrance (155 in fig 5 c)...' Or mention the line colour as you have done at other points. We revised the section in an attempt to make it clearer.

12. Figures 5 and 6: 'wind speed' should be 'wind velocity' as speed cannot be negative. This is now corrected.

13. Figure A1: adjust scales to the full model output can be seen (unless this is due to the model still stabilising during a spin-up period, in which case discard this data). Y-axis needs a label (local time or UTC?). Please see the discussion on the model spin-up time above and the new text in section 2.2 of the revised manuscript. We have revised Figure A1 to ensure that all the output can be seen and we have added labels to the y-axis.

**References (RC1)**

Bonekamp, P.N.J., Collier, E. and Immerzeel, W.W., 2018. The impact of spatial resolution, land use, and spinup time on resolving spatial precipitation patterns in the Himalayas. Journal of Hydrometeorology, 19(10), pp.1565-1581.

Jankov, I., Gallus Jr, W.A., Segal, M. and Koch, S.E., 2007. Influence of initial conditions on the WRF– ARW model QPF response to physical parameterization changes. Weather and Forecasting, 22(3), pp.501-519.

Román-Cascón, C., Steeneveld, G.J., Yagüe, C., Sastre, M., Arrillaga, J.A. and Maqueda, G., 2016. Forecasting radiation fog at climatologically contrasting sites: evaluation of statistical methods and WRF. Quarterly Journal of the Royal Meteorological Society, 142(695), pp.1048-1063.

Whiteman, C.D., 2000. Mountain meteorology: fundamentals and applications. Oxford University Press.

**RC2 – Anonymous Referee #2**

The manuscript titled "Daytime along-valley winds in the Himalayas as simulated by the WRF–model" is a good attempt to simulate winds particularly along the valleys in the complex terrain of Himalayan Mountains. In this work, four neighboring valleys are identified for the investigation on along-valley and cross-valley circulations in the Nepal Himalaya using the WRF model. Interestingly, point highlighted here is the interaction between local induced circulation and synoptic-scale flow, however, there are major concerns associated with methodology and persisting in the interpretation of the results as well.

**Major comments**

1. The five days period is too small for investigating the influence of synoptic scale flows on valley circulations, hence required to be explored to a longer time scale as strong seasonality may be associated with synoptic flows. The synoptic-scale flow are found to show pronounced variations from northerlies to the westerly. This could be due to the trough region embedded in the upper-level westerlies and moving eastward. If the longer period is considered, the overall variation of the synoptic-scale flow may change, and at some point, it can be southerly or southwesterly, which can strongly support the along-valley flows. Diurnality will also have a profound influence on the topographic flows due to thermal gradients. We would firstly like to stress that the main aim of the manuscript is not to understand the interaction between the valley winds and the synoptic-scale flow. We are sorry if our manuscript in the previous form gave this impression. We have now revised the introduction to ensure that our aims are clearly specified. Unfortunately, due to computational limitations, it is not feasible for us to perform long simulations covering the full annual cycle. Therefore, we chose to perform a case study during the post-monsoon season. We now provide justification for focusing on this season and for selecting these specific days in the new section 2.1. In addition, to understand better how climatologically representative the large-scale flow during our study period is, we have now compared the upper-level winds to the ERA5 climatology and present this new analysis in section 2.1 in the revised manuscript.

2. It is suggested to select a longer time period of at least two contrasting seasons. The valley inversions dominate during winter, and that might be confined within the valley region. The selected period is of winter, where the solar insulation is comparatively low, and the inversion can persist for a longer time. This can also affect along-valley flow. As stated above, computational and data storage limitations (storing the high temporal resolution data to see the temporal evolution of the winds require vast amount of disk space) mean that this is not possible. Our aim is to focus on the post-Monsoon / dry season only. We did not intentionally mean to give the impression that we present a full climatology of the thermally driven winds as this is clearly not possible from our 5-day simulation. We have carefully revised the manuscript (particularly the abstract, introduction and the new section 2.1) to ensure it is clear to a reader that we present a case study, not a climatological analysis. The solar irradiance at the NCO-P (Pyramid station in Khumbu

valley) is very similar during winter, the post-monsoon and monsoon season (Figure 4 in Bonasoni et al. 2010).

3. There are some issues with the model configurations and simulation. The spin-up time is not seen to be considered. The Land-surface scheme is not specified here, while it strongly impacts the near-surface fluxes and meteorological fields. We agree that the first 12 hours of the simulation should be not be considered for spin-up reasons. Text about this, including reference to the study by Bonekamp et al., (2018), has now been added to Section 2.2 (Section 2.1 in the first submission). We have added details of the land surface scheme to section 2.1. Figures 4 and 5 (Figures 3 and 4 in the first submission) have been updated by excluding the 17 Dec 2014 due to the spin-up period.

4. One of the major challenges to the models is to simulate the weak flow conditions. The local induced circulation is more prominent during the weak synoptic-scale flow (Solanki et al. 2019, BLM). It is strongly suggested here to investigate the model performance for two different wind flow regimes; low and strong. That can be performed only if the longer time period for simulations is taken into account. The aim of the manuscript is not to assess how well the model can represent the thermally driven winds under different large-scale conditions and therefore we feel this is out of scope of our current study. The main reason not to undertake a model evaluation study is that there are very very few observations available in this area and even those that are available are only basic meteorological measurements from automated weather stations. However, thank you for pointing out the study by Solanki et al (2019); we now refer to this at the start of section 4 of the revised manuscript.

5. A robust discussion was missing on how the selection of five days is made. This is a valid point. We now include a new subsection (section 2.1) explaining why we selected these days and also assess how representative these days are of the long term climate.

6. The wind follows the logarithmic variation within the surface layer. So it is suggested to compare the simulated winds to the observations at similar elevations (5m). Thank you for the suggestion. We have now done this assuming a neutral stratification and revised the comparison to observations with the adjusted height (Table 2 and Supplementary Figure A1). This adjustment of the modelled wind speeds improved the mean absolute errors from 0.6-3.8 m s$^{-1}$ to 0.4-2.5 m s$^{-1}$. In particular, the adjusted modelled daytime maximum wind speeds are closer to the observed than the unadjusted 10-m wind speeds.

7. The brief description on the observational data is not added here. In order to have close agreement, the comparison of the model output, preferably, is required to be made with the closest available instantaneous observations, along with mean values. Some text was previously provided in section 2.1.1 (now section 2.2.1). We have added some additional details about the observations to section 2.2.1.

8. Why is the altitude adjustment discarded? The actual altitude and model altitude comparison could be added to the description. The modelled 2-meter temperatures are now adjusted to correspond to the station altitude assuming a dry adiabatic lapse rate. The

height adjustment did not substantially improve the mean absolute errors. In addition, the height of the model grid points have now been added to Table 1.

9. Discuss the model evaluations using other matrices such as correlations, RMSE, and Mean Bias Error. In the comparison table, the MAE is much higher than the addressed benchmark values in literature, e.g., Emery et al., (2001). In addition, it would be interesting if the diurnal variation of wind and other meteorological variables, is presented. The WRF shows limited performance in simulating the diurnal cycle of winds over the Himalayas region (Singh et al., 2021). The Mean Bias Error and Root Mean Square Error have now been added to the comparison of the modelled fields to the observations. We have revised Supplementary Figure A1 to now show the modelled wind speed adjusted to 5 m rather than the 10-m wind. In addition, Figure A1 now also shows the 2-meter temperatures that have been adjusted for differences between the height of model grid points and the observation stations.

10. Line 176, the decomposition of the vector A, into along- and cross-valley is carried out taking the three-points on either sides of valley center, to avoid the sharp gradients. However, the valley is narrowing towards the north, so the taking six points for the actual orientation of the valley, will change largely, and hence may not be a close representation. Here it is suggested to compare these two components, along-valley, and cross-valley, by choosing one-, two- and three points. If there is not large variability between one- and three-, the orientation of the valley may be considered to be represented correctly. Thank you for the suggestion. We have now performed the suggested analysis and it is shown here in the response in Figures 1 and 2. Figure 1 is the same as Figure 6 in the updated manuscript (along-valley wind timeseries) but without the 3 grid point smoothing of the valley center line. Figure 2 is the same as Figure 6 in the updated manuscript (along-valley wind timeseries) but without the 3 grid point smoothing of the valley center line or the 5 grid point averaging of the along-valley wind speeds around the marked grid point. The outcome is nearly the same as in Figure 6 in the manuscript, but the outcome is still more noisy. We suggest to use the 3 grid points smoothing of the valley center line and the 5 grid point averaging of the along-valley wind for each grid point for better representation of the along-valley winds.

11. The investigation of the cross-valley component is not discussed in detail. The cross-section of the valleys for 19-21 December showing the intrusion of the synoptic-scale flow into the valley region is to be presented/plotted. Main focus of the study is in the daytime along-valley winds as the manuscript title implies. Cross-valley cross-sections have now been added as a new Supplementary Figures A3–A4 in the revised manuscript.

**Specific comments**

1. In figure 2, there are some visible artifacts (locations of about 1000m high, as per the scale) on the topography data, along the costal lines extending from Arabian Sea to Bay of Bengal. Topography needs to be checked. We are not 100% sure what this comment refers to. The topography plotted in color in Figure 1 (now Figure 2) is directly the

surface height taken from the WRF model output (as stated in the caption). We checked this was plotted correctly and it appeared to be correct. However, we think the features the reviewer is referring to are relatively small and also a very long way away from the high-resolution study area.

2. Line 126-128: The local circulations change with the day-night contrast including two transitions during sunset and sunrise. Better to select the stable atmosphere during night-time and convective daytime hours by discarding such transitions. This refers to the night and day periods the skills scores were computed over. The transition is relatively quick though and we do not want to exclude these time periods as (1) they are interesting and (2) because we do not want to reduce the amount of data points in these comparisons.

3. Line 144-145: The observed wind speed is below $2\,\mathrm{ms}^{-1}$, while the corresponding MAE is $2.1\,\mathrm{ms}^{-1}$. The model performance is not very convincing for the low wind conditions. In the first submission these MAE were calculated comparing 5 meter observed and 10 m modelled wind, in the updated comparison the error is smaller. In addition, our RMSE values for both wind and temperature are notably smaller than those presented by Bonekamp et al (2018) - we add some discussion about this to section 2.2.1 of the revised manuscript.

4. Line 163-164: it would be interesting to see a close view of the individual valley, (possibly 3-d view to have a better understanding of the orientation and shapes) that will highlight the geometrical differences between the valleys. Thank you for the suggestion. We did try to produce 3D images but they were very messy and difficult to understand on a 2D screen / piece of paper. Therefore, we concluded that the current 2D-topography map, in addition with the along-valley profiles, are adequate for a reader to gain a good understand of the valley shapes.

5. Line 166: How is this criterion selected? The valley width is calculated at an elevation of 1000 m above the valley center line. The elevation of 1000 m above the valley center line is based on the depths, most of the valleys have a minimum depth of 1000 m from the bottom of the valleys. We have revised the last paragraph of the section 2.3.1 (2.2.1 in the first submission) to make it more clear.

6. Line 226: Is it the pressure level corresponding to the ridge height? What is the criterion of this selection? Why not 500hPa, which is generally used in synoptic weather charts? We analysed 400 hPa instead of the more commonly used 500 hPa as in this region some peaks have a surface pressure less than 500 hPa and hence this pressure surface is below the surface. We therefore used 400 hPa to be above the topography but not too far above the ridges. The reason for selecting 400 hPa over 500 hPa is now included also in the updated manuscript in section 2.1.

7. Line 226-228: this discussion may change for the longer run of the model. As we were unable to perform a longer model run, this discussion has not changed

8. Line 242: This is not a sufficient explanation for neglecting this period here. The inter-action between two flows should be investigated separately. This refers to "Due to the

interruption of the thermally driven winds by the large-scale winds on 17-18th Dec, the analysis mostly concentrates on the 20–21st Dec in the following sections". As our aim was to investigate the along-valley winds under the simpliest situation - when there is no interaction with the large-scale - we do not investigate how the two flows interact with each other. We have revised the introduction and abstract to make the aim of the paper clearer and in addition have revised this specific sentence to re-iterate that the focus is not on the interaction with the large-scale flow.

9. Line 243: During 19-21 Dec, the flow is reported to be westerlies, while it can still impact the thermal winds in the valleys like northerly, depending upon the orientation of the valleys and magnitude of the westerlies. How can the interruption of the thermal winds by synoptic westerlies be neglected? We cannot be 100% certain that the synoptic-scale flow does not influence the thermally driven along-valley winds on the 19 - 21 December. However, the modelled wind speeds and directions strongly suggest that these winds are caused by the valley wind mechanism: there is a clear diurnal cycle with the expected wind directions and the wind speeds are lower than during the 17 Dec when there was a clear synoptic-scale influence. In addition, we did study cross sections of vertical velocity (not shown) and did not seem any evidence of the larger-scale winds penetrating down into the valleys.

10. Figure 2: Why the valley width is missing at some grid points? For some valleys, in the southern parts where the valleys are less deep, the ridge height may be less than 1000 meter higher than the valley center line height (i.e. the depth of the valley is less than 1000 m) on either or both slopes. For these along-valley grid points on the valley center lines the valley width can not be determined using this method.

11. Line 246: Is it thermal-induced local circulation or synoptic flow? Moreover, it is unlikely characteristic of mountain meteorology, if it is happening so (evening time upslope winds), then meteorological processes operating at high altitudes are required to be investigated with specific observational experiments. This refers to this statement: "Up to 25 m s$^{-1}$ near-surface winds are found on the ridges surrounding the Kanchanjunga valley, but within the valley atmosphere the near-surface winds stay below 10 m s$^{-1}$ and flow in the up-valley direction." Therefore, the winds on the ridges are synoptic-scale flow but the winds in the valleys are the thermally driven winds which are weaker and flowing in the up-valley direction (as expected during the day).

12. Figure 4: Remove the extra contour lines related to the topography. Due to the jumbling contour lines, the wind field is not very clear. We attempted to improve the clarity of this figure. The same color scheme as used in the Figure 4 (400–hPa winds in the outer domain d01 – Figure 3 in the first submission) is now used in this Figure 5 (25m winds in inner domain - Figure 4 in the first submission). The number of topography contours has also been reduced to two (3000 and 5000 masl) instead of contours every 1000m.

13. Line 265-266: How the diurnal cycle is explained by taking a single hour. If the northerlies prevailed during 17-18 Dec, it would remain unchanged irrespective of the hour. This

statement refers to Figure 6 which shows the full timeseries. To make it clearer, we have added another reference to Figure 6.

14. Line 275-276: Show the observation and model comparison for these calm wind conditions. It is very interesting if the model and observations are in agreement. This comparison was presented in the supplementary material Figure A1 in the first submission and is now updated with the height adjustments in 5m wind and 2m temperature.

15. Figure 5: The scale on y-axis is not correct. Many of the points are out of the scale and not visible. y-scale of the Figure 6 (Figure 5 in the first submission) is now fixed to show the daytime maximum values in panel a. New Supplementary Figure A2 added in the manuscript which is the same as Figure 6 but with extended y-scale to show the minimum along-valley wind velocities in panels a and c.

16. Line 333: But not all valleys are north-south oriented. These model levels can be well above the near valley inversions during the nighttime. There appears to be two separate issues in this comment. First, we agree that not all parts of all valley are north-south oriented but as described in section 2.2.2, we account for this fact by computing the orientation of the valley at each point. The second issue concerns the height of model levels and we do agree that the lowest model level at 25 m a.g.l may be above the inversion at night. We have revised this sentence to now read "...may not capture the thermally driven cross-valley circulation, due to the shallower nature of slope winds and the fact that nighttime inversions may be below the lowest model level".

17. Line 366-367: Suggested to add a figure on valley cross-section wind. Cross-valley cross-sections added as a new Supplementary Figures A3–A4 in the manuscript. Panels a, e and g show the locations of maximum daytime along-valley wind speed referred in this comment.

**References (authors response)**

Bonasoni, P., Laj, P., Marinoni, A., Sprenger, M., Angelini, F., Arduini, J., Bonafè, U., Calzolari, F., Colombo, T., Decesari, S., Di Biagio, C., di Sarra, A. G., Evangelisti, F., Duchi, R., Facchini, M., Fuzzi, S., Gobbi, G. P., Maione, M., Panday, A., Roccato, F., Sellegri, K., Venzac, H., Verza, G., Villani, P., Vuillermoz, E., and Cristofanelli, P.: Atmospheric Brown Clouds in the Himalayas: first two years of continuous observations at the Nepal Climate Observatory-Pyramid (5079 m), Atmospheric Chemistry and Physics, 10, 7515–7531, `https://doi.org/10.5194/acp-10-7515-2010`, `https://acp.copernicus.org/articles/10/7515/2010/`, 2010.

Bonekamp, P., Collier, E., and Immerzeel, W.: The Impact of Spatial Resolution, Land Use, and Spinup Time on Resolving Spatial Precipitation Patterns in the Himalayas, Journal of Hydrometeorology, 19, `https://doi.org/10.1175/JHM-D-17-0212.1`, 2018.

[Figure]

Figure 1: Along-valley wind timeseries. Figure 6 in the manuscript but without the 3 grid point smoothing of the valley center line when calculating the along-valley wind component.

[Figure]

Figure 2: Along-valley wind timeseries. Figure 6 in the manuscript but without the 3 grid point smoothing of the valley center line and also without the 5 neighbouring grid point along-valley averaging when calculating the along-valley wind component.

---

## Referee Report (RR1)

Thanks to the authors for the new addition of section 2.1, which clearly justifies the choice of this time period for analysis. Both this and the alterations to the text make the aims of the paper clearer. I also found the new cross-sectional diagrams in A3 and A4 very interesting, especially given the lack of 'textbook' cross-valley circulation at many locations.

I have a few small comments:

1. As I understand from line 111, the 17th December is now removed from all analysis, therefore the manuscript should refer to a 4-day period, rather than a 5-day period.
2. As model spin-up is a computational artifact, it is standard practice not to show it in results and therefore I'd recommend it's removed from all graphs in the manuscript (line 535). As the 17th December is not discussed in the analysis, I suggest it is removed from all graphs entirely for clarity.
3. Conclusions: As 17th December is not discussed in the results, discussion of this day should be removed from the conclusions (line 594: only north-westerlies have been discussed on the 18th December in the results).
4. Figures A3 and A4: these are very interesting, thank you for the addition! As the numbers are a little small, an addition in the figure caption of 'solid lines are positive values and dashed lines are negative values' would be helpful for clarity.
5. 91 typo: coincided -> coincide
6. 140: As using a spin-up period is standard practice in numerical modelling, this paragraph could be shortened to just state the spin-up period and reference Bonekamp 2018.
7. 211: Typo, repetition of 'WRF-simulation'
8. Figure 7: I would recommend adding that the wind velocity has been multiplied by -1 (e.g. m s$^{-1}$ (multiplied by -1)) to the y-axis label on the western slope, to make sure this important point is not missed.

---

## Referee Report (RR2)

The authors have resolved some major issues related to the methodology and interpretation of the results to meet the main objectives of this manuscript. While the following issues still persist:

**Major comments**

1. The explanation for the longer model run and spin-up period (12 h) is acceptable for such a case study as the objective of this study is to investigate the local valley circulations that are induced by topography and profoundly affected by variations in the synoptic-scale flows. However, the synoptic scale wind flow in Figure 1 is shown to vary enormously throughout the period and wind patterns significantly changed from one day to the other (17-18 December) and are not rather appropriate for the analysis, in order to get the true representation of valley flows. It would have been the best choice to look into the days-in-continuation where the synoptic flows are not changing to a greater extent i.e. few days before and after the slight change, to ensure that synoptic flows are not random but systematic. Since the variable synoptic scale flows appear to affect the intensity of local valley circulations, which could be investigated separately.

2. The cross valley circulations are primarily originating as a result of anabatic and katabatic flows and dominant during the daytime in the weak mean flow conditions. This could be a better way to investigate and define the cross valley circulations and differentiate among them. Therefore, the justification for selecting this period is not sufficient. If similar synoptic flow conditions are chosen, then the differences in the day-night valley circulations will be only due to the thermally driven processes.

3. To be more specific, as evident from Figure 1(discarding 17-18 Dec) that after 21 December for the next few days, the synoptic-scale flow does not change much. The selection of this period will support the third main aim of this study which is related to investigating the cause of the differences in local winds. Further, the influence of the synoptic-scale flow (e.g., 15 to 18 December when the wind direction changes) on along-valley flow will add more value. This work is really good contribution to the mountain meteorological studies.

**Specific comment:**

1. Line 152-153: It is suggested to quote the name of the valleys in Figure 2b.

---

## Author Response (AR2)

**Response to Referees – Daytime along-valley winds in the Himalayas as simulated by the WRF–model**

Johannes Mikkola, Victoria Sinclair, Marja Bister, Federico Bianchi

November 2022

We thank the reviewers for their constructive comments on our submitted manuscript. We have copied the comments of the reviewers in black here and include our response to each individual comment in blue. In the revised submission the Supplementary Figures A1–A4 have been re-labeled as Supplementary Figures S1–S4.

**Referee report 1**

Thanks to the authors for the new addition of section 2.1, which clearly justifies the choice of this time period for analysis. Both this and the alterations to the text make the aims of the paper clearer. I also found the new cross-sectional diagrams in A3 and A4 very interesting, especially given the lack of 'textbook' cross-valley circulation at many locations.

I have a few small comments:

1. As I understand from line 111, the 17 th December is now removed from all analysis, therefore the manuscript should refer to a 4-day period, rather than a 5-day period. Since we consider the time period from 18LT on the 17th December, the wording of "5–day period" was used in the previous revision as technically there are 5 different dates. However, in the revised manuscript we have changed this to a "4–day period" since we only consider 4-day time periods (i.e. the white areas in Figure 6), that are the main aim of the study.

2. As model spin-up is a computational artifact, it is standard practice not to show it in results and therefore I'd recommend it's removed from all graphs in the manuscript (line 535). As the 17 th December is not discussed in the analysis, I suggest it is removed from all graphs entirely for clarity. Thank you for the suggestion. For clarity, the spin-up period is excluded in all of the timeseries figures.

3. Conclusions: As 17 th December is not discussed in the results, discussion of this day should be removed from the conclusions (line 594: only north-westerlies have been discussed on the 18 th December in the results). This sentence referred to by the reviewer discussed the northerlies and north-westerlies during the remaining six hours of the 17th (after the spin-up 06-18 local time) and the early morning hours of the 18th. We have carefully revised the text to make sure the period of model spin-up is not considered in the conclusions.

4. Figures A3 and A4: these are very interesting, thank you for the addition! As the numbers are a little small, an addition in the figure caption of 'solid lines are positive values and dashed lines are negative values' would be helpful for clarity. Thank you, the figure caption is now revised.

5. 91 typo: coincided → coincide This is now corrected.

6. 140: As using a spin-up period is standard practice in numerical modelling, this paragraph could be shortened to just state the spin-up period and reference Bonekamp 2018. Based on the referee comments on the first submission, we feel the first general sentence about model spin-up is relevant. Also there is only one sentence that could be deleted and we think the paragraph is not too long as it is. Therefore, we have not made any revision in the manuscript concerning this point.

7. 211: Typo, repetition of 'WRF-simulation' This is now corrected.

8. Figure 7: I would recommend adding that the wind velocity has been multiplied by -1 (e.g. m s $^{-1}$ (multiplied by -1)) to the y-axis label on the western slope, to make sure this important point is not missed. We added text boxes below each column in this figure stating that the positive (negative) values refer to up-slope (down-slope) winds. We think this is clearer, and less confusing, than adding information on the y-axis of the panels showing the western slope.

**Referee report 2**

The authors have resolved some major issues related to the methodology and interpretation of the results to meet the main objectives of this manuscript.

We thank the reviewer for valuable ideas how to extend the range of the wind analysis and build credibility of our study. Here we gather first a general response on their main points and give specific comments below.

Our current study is the first step in understanding thermally driven winds in the Himalayas and also the transport of aerosol by these winds. As a starting point, in this study, we focus heavily on the valley topography driven differences in the daytime up-valley winds. Unfortunately, the simulation used in this study under review was performed on our super computing system quite a long time ago and since then major system updates have occurred. This means that the exact model configuration is currently not available and hence it is very difficult for us to extend this simulation. We do however fully acknowledge that due to our short time period our conclusions are subject to uncertainties and we have carefully revised the manuscript to state this. Finally, we would state that in our ongoing and future work we are attempting to resolve some of these uncertainties. Currently, we are studying thermally driven winds and aerosol transport processes at the same location in the Himalayas by means of a 1-month long high-resolution simulation with fully coupled meteorology and air-chemistry. However, the results from this longer simulations constitute future work and are beyond the scope of this current paper.

While the following issues still persist:

**Major comments**

1. The explanation for the longer model run and spin-up period (12 h) is acceptable for such a case study as the objective of this study is to investigate the local valley circulations that are induced by topography and profoundly affected by variations in the synoptic-scale flows. However, the synoptic scale wind flow in Figure 1 is shown to vary enormously throughout the period and wind patterns significantly changed from one day to the other (17-18 December) and are not rather appropriate for the analysis, in order to get the true representation of valley flows. It would have been the best choice to look into the days-in-continuation where the synoptic flows are not changing to a greater extent i.e. few days before and after the slight change, to ensure that synoptic flows are not random but systematic. Since the variable synoptic scale flows appear to affect the intensity of local valley circulations, which could be investigated separately. Figure 1 shows the synoptic scale winds in the outer domain of the simulation. The actual study area is in the North-Eastern part of Nepal (Nepal borders are marked on red in the Figure 1). If only the wind in this smaller area is considered, we think it is somewhat of an overestimation to state that the synoptic-scale wind varies enormously during the study period 18-21 Dec. We do agree that the synoptic-scale wind differs on the 17th but we do not consider this day in our analysis. However, as stated above, we have revised the manuscript to be clear that our results are subject to uncertainties and we state that as future work, longer simulations

are needed to fully assess the impact of the synoptic-scale flow on the thermally driven winds.

2. The cross valley circulations are primarily originating as a result of anabatic and katabatic flows and dominant during the daytime in the weak mean flow conditions. This could be a better way to investigate and define the cross valley circulations and differentiate among them. Therefore, the justification for selecting this period is not sufficient. If similar synoptic flow conditions are chosen, then the differences in the day-night valley circulations will be only due to the thermally driven processes. In this study we focus on the daytime along-valley winds and therefore the cross-valley winds are not analysed and discussed in depth in the manuscript. Detailed analysis of the cross-valley winds will be included in the future work.

3. To be more specific, as evident from Figure 1(discarding 17-18 Dec) that after 21 December for the next few days, the synoptic-scale flow does not change much. The selection of this period will support the third main aim of this study which is related to investigating the cause of the differences in local winds. Further, the influence of the synoptic-scale flow (e.g., 15 to 18 December when the wind direction changes) on along-valley flow will add more value. This work is really good contribution to the mountain meteorological studies. The combined effect of the synoptic scale flow and the valley topography on the local along-valley winds would be indeed interesting and as stated above, we plan in the future to perform longer simulations, with coupled chemistry, to better address this question.

**Specific comment:**

1. Line 152-153: It is suggested to quote the name of the valleys in Figure 2b. This is now added in the figure caption.

---

## Author Response (AR3)

**Response to the Editor – Daytime along-valley winds in the Himalayas as simulated by the WRF–model**

Johannes Mikkola, Victoria Sinclair, Marja Bister, Federico Bianchi

November 2022

We thank the Editor for their constructive help throughout the review-process of the manuscript.

We made the suggested small changes to the abstract and in the last section of the manuscript.